# A Phonological Study of Rongpa Choyul

## Jingyao Zheng

Laboratory of Language Sciences, Peking University, Beijing 100871, China; Choyulskad@pku.org.cn

**Abstract:** This paper presents a detailed description of the phonology of the Rongpa variety of Choyul, an understudied Tibeto-Burman language spoken in Lithang (理塘) County, Dkarmdzes (甘孜) Tibetan Autonomous Prefecture of Sichuan Province, China. Based on firsthand fieldwork data, this paper lays out Rongpa phonology with details, examining its syllable canon, initial and rhyme systems, and word prosody. Peculiar characteristics of this phonological system are as follows: First, Rongpa has a substantial phonemic inventory, which comprises 43 consonants, 13 vowels, and 2 tones. 84 consonant clusters are observed to serve as the initial of a syllable. Secondly, the phonemic contrast between plain and uvularized vowels is attested. In addition, regressive vowel harmony on uvularization, height, and lip-roundedness can be clearly observed in various constructions including prefixed verb stems. Finally, regarding word prosody, two tones in monosyllabic words, /H/ and /L/, are observed to distinguish lexical meanings, and disyllabic words exhibit four surface pitch patterns. Pitch patterns in verb morphology are also examined. The findings and analyses as presented in this paper could form a foundation for future research on Rongpa Choyul.

**Keywords:** Tibeto-Burman; Choyul; Rongpa; phonology

## 1. Introduction

Choyul (a.k.a. Choyu, Chuyu, Queyu, and, in Chinese, Quèyù, 却域) is an understudied Tibeto-Burman language spoken primarily by thousands of people residing along Nyagchu River (ཉག་ཆུ Yǎlóngjiāng 雅砻江), primarily in Dkarmdzes (དཀར་མཛེས Gānzī 甘孜) Tibetan Autonomous Prefecture, Sichuan Province, China, specifically in the counties of Nyagchu (ཉག་ཆུ Yǎjiāng 雅江), Rta'u (རྟའུ Dàofú 道孚), Nyagrong (ཉག་རོང Xīnlóng 新龙), and Lithang (ལི་ཐང Lǐtáng 理塘) (Lu 1985). It is also reported to be spoken in the hamlet of Thabsmkhas (ཐབས་མཁས Tǎgé 塔格) of Lhasgang (ལྷ་སྒང Tǎgōng 塔公) Township, Darmdo (དར་མདོ Kāngdìng 康定) County (Suzuki and Wangmo 2016, 2018).

Previous studies on Choyul are scanty. The earliest linguistic description dates back to Lu (1985), which is a sketch of the phonology and grammar of the speech form spoken in Tuanjie Township (团结乡), Nyagchu County.[1] Wang (1991) presents a language profile of the variety of gYanglagshis (གཡང་ལག་གཤིས Yóulāxī 尤拉西) in Nyagrong County. Nishida (2008) published a preliminary phonological system of Rongpa Choyul, based on the speech form of bTsanmda Hamlet (བཙན་མདའ Zēngdá cūn 增达村). Suzuki et al. (manuscript) presents a phonological study of Lhasgang Choyul, and Suzuki and Wangmo (2016) also lay out the sociolinguistic status of Lhasgang Choyul. Suzuki and Wangmo (2018) provide a wordlist of Lhasgang Choyul, which contains around 750 words with their equivalents in the neighboring Tibetan language. Xuan Guan, a PhD candidate in the University of Oregon, has been investigating the Pubarong (ཕུ་བ་རོང Pǔbāróng 普巴绒) variety of Choyul for several years. Her research findings were presented in several conferences over the past few years. Figure 1 below illustrates the geographical locations where varieties of Choyul have been documented to this date.

The term "Choyul" is translated as "Quèyù (却域)" in Chinese. Wang (1991, p. 46) mentions that "yù (域)" may be etymologically related to the Tibetan morpheme "ཡུལ <yul>", meaning "region; place", but the origin and meaning of "què" are yet to be figured out. One

possibility is that it refers to the regions alongside the Nyagchu river, where this language is spoken (Wang 1991, p. 46; Suzuki 2018).[2]

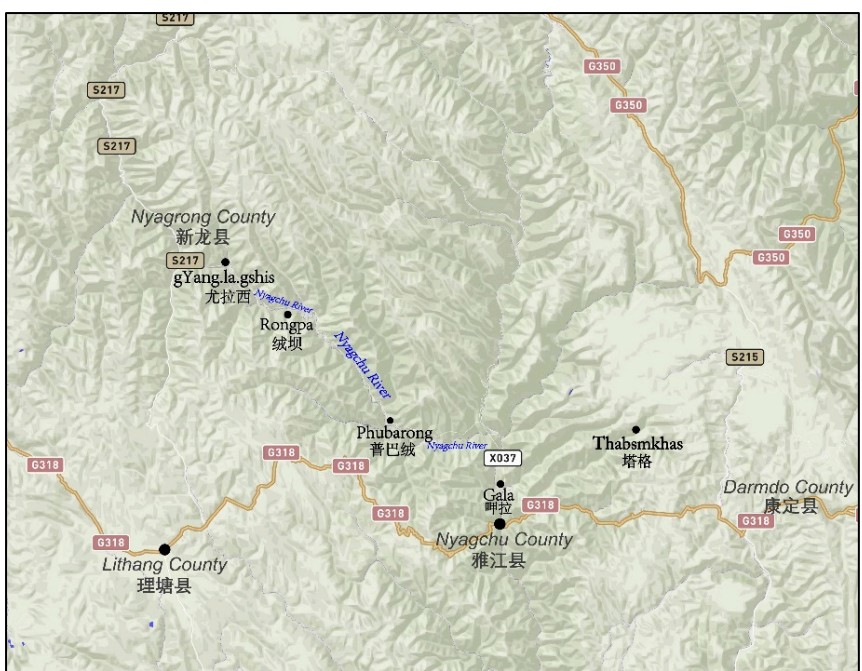

**Figure 1.** The geographical location of Choyul varieties.

The genetic affiliation of Choyul is yet to be determined. It was tentatively assigned to the "Qiangic" branch by previous scholars (Lu 1985[3]; Wang 1991; Sun 1983, 1991, 2001, 2016). Lu (1985, p. 67) stated that Choyul is closely related to Qiang and Pumi, and thus it should be classified as affiliated to the Qiangic branch. Choyul was once considered as a member of the southern branch of Qiangic (Sun 1983, 2001), but was afterwards subsumed into the central branch (Sun 2016, p. 4).[4] However, Chirkova (2012) has argued that it could be more adequate to regard "Qiangic" as an areal grouping rather than a genetic subgroup of Sino-Tibetan. She argues that the "Qiangic subgroup" is controversial for the following four reasons: (1) common typological features are also found in other neighboring non-Qiangic languages, which is considered as the result of parallel developments or areal phenomena; (2) the small proportion of shared common vocabulary; (3) the lack of common innovations; and (4) the historical, ethnic, and linguistic complexity of the Qiangic languages area. Thus, the genealogical relationships among "Qiangic" languages remain to be discussed, and the genetic position of Choyul is yet to be finalized.

The speech form examined in this paper, the Rongpa variety of Choyul (henceforth Rongpa), is spoken by people from the townships of Rongpa (རོང་པ Róngbà 绒坝, in Rongpa /roʁL peH/), Lhayul (ལྷ་ཡུལ Hāyī 哈衣, in Rongpa /xkəH mdiH/), and Dga'khog (དགའ་ཁོག Gākē 呷柯, in Rongpa /gəL khəH/) in Lithang County. Rongpa speakers are identified ethnically as Tibetan by the official government and traditionally follow common Tibetan cultural practices and practice Tibetan Buddhism. Over the past decade, Rongpa speakers have been relocated from their home villages to Lithang County due to the construction of water dams and hydropower stations. They often refer to their language as /roL xkɜH/ (རོང་སྐད <rong.skad>), meaning "the speech of the farming area (in Mandarin Chinese nóngqūhuà 农区话)" (Suzuki 2018), which is not mutually intelligible with the neighboring Tibetan dialects, namely the dialect of the nomadic area (/ɣaʁL xkɜH/ ཝ་སྐད <wa.skad> mùchǎnghuà 牧场话) and the dialect of Lithang (/poL xkɜH/ བོད་སྐད <bod.skad> zànghuà 藏话, which literally means "Tibetan"), the latter of which is spoken by up to three-quarters of the population in Lithang County. Other than Tibetan and Choyul, Sichuan Mandarin (/dʒɛL xkɜH/

ক্রুঝ <rgya.skad> hànhuà 汉话, which literally means "Chinese") is also commonly used in the area.

The phonological analysis presented herein is based on materials collected during several recent fieldtrips to Gaocheng Town (gāochéngzhèn 高城镇) in Lithang County in 2021 and 2022. The main dataset is a word list that contains approximately 2000 entries. Each entry was recorded with five repetitions and spoken in isolation using the digital recorder (SONY ICD-UX575F) at a sampling rate of 48 kHz. Recordings were saved as .wav files and analyzed using Praat (Boersma and Weenink 2018). We also elicited the vowel alternation patterns of a hundred verbs that index the person and number of the subject. Our main consultant is Ms. Tshelhamo (ཚེ་ལྷ་མོ 泽拉姆), 43 years old in 2023. She was born in Rimda' Village (རི་མདའ <ri.mda'> Réndá cūn 仁达村), Rongpa Township. She lived in Rimda until 2016; that year, she and the other villagers were relocated to Gaocheng Town by the local government. In addition to Choyul, she also has a good command of the neighboring Tibetan dialects and Sichuan Mandarin.

The remainder of this paper starts with a description of the syllable structure of Rongpa in Section 2. Section 3 discusses the initial inventory that comprises simple consonant initials (Section 3.1) and complex initials (Section 3.2). Then, in Section 4, the vocalic system, including plain vowels (Section 4.1) and uvularized vowels (Section 4.2), will be analyzed, followed by a brief discussion of vowel harmony alternations (Section 4.3). Section 5 focuses on the suprasegmental phenomena of Rongpa. As a conclusion, the main findings will be summarized in Section 6.

## 2. Syllable Canon

The syllable canon of Rongpa is quite simple, and can be formulated as:

$$(C)(C)(C)V$$

All Rongpa syllables are open. The optional onset is composed of, at most, three consonant slots, followed by an obligatory simple vowel slot. The minimum syllable is a single vowel, e.g., /$\varepsilon^H$/ 'question marker'. The simple initial slot C allows all 43 consonant phonemes.

Complex $C_1C_2$ initials abound, which can be divided into 4 subtypes in terms of the member that appears in the first slot, $C_1$. $C_2$ that are compatible with each type of $C_1$ initial are also summarized in Table 1:

**Table 1.** Rongpa $C_1C_2$ clusters as complex initials.

| | First Element ($C_1$) | | Second Element ($C_2$) |
|---|---|---|---|
| a. | dorsal fricative | /x/ /ɣ/ | obstruents; voiced nasals; voiced lateral; |
| b. | nasal | /m/ /N/[5] | voiceless unaspirated stops and affricates; voiced stops and affricates; voiced nasal /n/; |
| c. | bilabial stop | /p/ /b/ | all coronal consonants; labiodental fricative /v/; |
| d. | velar consonant | /k/ /kʰ/ /ɡ/ | labial glide /w/ |

The maximum syllable is CCCV. Note that in Rongpa, only /xkw/ is observed, e.g., /xkwɛ$^H$/ 'be shy'; /xkwa$^{ɢL}$/ 'to pick: 3'.

Table 2 below exhibits all the syllable types that have been attested in Rongpa.

**Table 2.** Rongpa syllable types.

| Syllable Types | Examples | |
|---|---|---|
| V | /ɛ$^H$/ 'question marker'[6] | /a$^ʁ$$^H$ tsa$^ʁ$$^L$/ 'turnip' |
| CV | /mu$^H$/ 'sky' | /sə$^L$/ 'nit' |
| C$_1$C$_2$V<br>C$_1$=/x/ or /ɣ/ | /xpa$^ʁ$$^L$/ 'toad' | /ɣma$^ʁ$$^H$/ 'bamboo' |
| C$_1$=/m/ or /N/ | /mdʑi$^H$/ 'thunder' | /Nt$^h$e$^H$/ 'to sing' |
| C$_1$=/p/ or /b/ | /ptʂɛ$^L$/ 'friend' | /bzi$^H$/ 'marmot' |
| C$_1$=/k/, /k$^h$/ or /g/ | /k$^h$wa$^ʁ$$^L$/ 'to pick: 3' | /gwɛ$^L$/ 'to block: 3'[7] |
| C$_1$C$_2$C$_3$V | /xkwɛ$^L$/ 'charcoal' | /xkwa$^ʁ$$^H$/ 'to carve: 3' |

## 3. Onsets

The onset system of Rongpa is quite complex, with 43 simple initial consonants (Section 3.1) and at least 84 complex initials (Section 3.2).

### 3.1. Simple Consonant Initials

Table 3 presents the 43 simple consonant initials, all of which can occur in the C slot as the onset of CV syllables.

**Table 3.** Rongpa simple consonant inventory (consonants enclosed in parentheses are phonemes, each of which is only observed in one example). (43 consonants).

| Manner | | Place | Labial | Alveolar | Postalveolar | Retroflex | Palatal | Velar |
|---|---|---|---|---|---|---|---|---|
| plosive | voiceless | −ASP | p | t | | | | k |
| | | +ASP | p$^h$ | t$^h$ | | | | k$^h$ |
| | voiced | | b | d | | | | ɡ |
| affricate | voiceless | −ASP | | ts | tʃ | tʂ | | |
| | | +ASP | | ts$^h$ | tʃ$^h$ | tʂ$^h$ | | |
| | voiced | | | dz | dʒ | dʐ | | |
| nasal | voiceless | | m̥ | n̥ | | | | (ŋ̊) |
| | voiced | | m | n | | | | ŋ |
| trill | voiceless | | | r̥ | | | | |
| | voiced | | | r | | | | |
| fricative | voiceless | −ASP | (f) | s | ʃ | ʂ | | x |
| | | +ASP | | s$^h$ | ʃ$^h$ | ʂ$^h$ | | |
| | voiced | | v | z | ʒ | ʐ | | ɣ |
| approximant | voiced | | w | | | | j | |
| lateral approximant | voiceless | | | l̥ | | | | |
| | voiced | | | l | | | | |

Below are some salient characteristics of the simple consonantal initials.

1.  Rongpa obstruents (i.e., plosives, affricates, and fricatives) show a three-way contrast between voiceless unaspirated, voiceless aspirated, and voiced. It is noteworthy that Rongpa has contrastive aspirated fricatives, which is rather rare cross-linguistically. The minimal or near-minimal triplets in (1) illustrate the distinctions:[8]

(1)  /p/   /**po**$^H$/ 'to dig: 1sg'        /t/    /**to**$^{ʁH}$/ 'wing'
     /pʰ/  /**pʰ**o$^H$/ 'to worship: 1sg'   /tʰ/   /**tʰ**o$^{ʁH}$/ 'meat; flesh'
     /b/   /**bu**$^H$/ 'breath'             /d/    /**do**$^{ʁL}$/ 'to saw: 1sg'

     /ts/  /**tso**$^H$/ 'to sit: 1sg'       /ʧ/    /**ʧ**o$^L$/ 'to wear (ring): 1sg'
     /tsʰ/ /**tsʰ**o$^H$/ 'to dye: 1sg'      /ʧʰ/   /**ʧʰ**o$^H$/ 'to eat: 1sg'
     /dz/  /**dzo**$^H$/ 'to teach: 1sg'     /ʤ/    /**ʤ**o$^H$/ 'to run: 1sg'

     /tʂ/  /**tʂ**ə$^H$/ 'this'              /k/    /**k**ə$^H$/ 'age'
     /tʂʰ/ /**tʂʰ**ə$^H$ Nbu$^H$/ 'buttocks' /kʰ/   /**kʰ**ə$^H$/ 'to need'
     /dʐ/  /**dʐ**ə$^H$/ 'to ruminate (animals)' /g/ /**g**ə$^H$/ 'vulture'

     /s/   /**se**$^H$ Nbe$^H$/ 'cotton'     /ʃ/    /**ʃ**ə$^H$/ 'iron spoon'
     /sʰ/  /**sʰ**e$^L$/ 'firewood'          /ʃʰ/   /**ʃʰ**ə$^H$/ 'worship'
     /z/   /**ze**$^L$/ 'liver'              /ʒ/    /**ʒ**ə$^H$/ 'to move'

     /ʂ/   /**ʂ**ɜ$^L$ wu$^H$/ 'weed'
     /ʂʰ/  /**ʂʰ**ɜ$^L$ wu$^H$/ 'money'
     /ʐ/   /**ʐ**ɜ$^L$/ 'travel-ready dishes'

2. Affricates and fricatives show a three-way contrast in their places of articulation (i.e., alveolar, postalveolar, and retroflex):

(2)  /ts/  /**ts**ə$^H$/ 'to squeeze: 1pl'   /tsʰ/  /**tsʰ**ə$^{ʁH}$/ 'thorn'
     /ʧ/   /**ʧ**ə$^L$ ʧə$^H$ pa$^{ʁL}$/ 'cicada' /ʧʰ/ /**ʧʰ**ə$^{ʁL}$ ʧʰə$^{ʁH}$/ 'sweet'
     /tʂ/  /**tʂ**ə$^H$/ 'this'              /tʂʰ/  /**tʂʰ**ə$^{ʁL}$ tʂʰə$^{ʁH}$/ 'be pleasant'

     /s/   /**s**ə$^L$/ 'nit'                /dz/   /**dz**ɛ$^H$/ 'to teach: 1pl'
     /ʃ/   /**ʃ**ə$^H$/ 'cypress'           /ʤ/    /**ʤ**ɛ$^H$/ 'to run: 1pl'
     /ʂ/   /**ʂ**ə$^H$/ 'louse'             /dʐ/   /**dʐ**ɛ$^H$/ 'enemy'

3. Voiced nasals and liquids show contrasts with their voiceless counterparts, though the occurrences of voiceless nasals and liquids are rare, as minimal pairs in (3) show:

(3)  /m/   /**ma**$^{ʁH}$ xka$^{ʁL}$/ 'rainbow'  /n/  /**ne**$^L$/ 'you'
     /m̥/   /**m̥**a$^{ʁL}$/ 'eared pheasant'     /n̥/  /**n̥**e$^H$ n̥e$^L$/ 'red'

     /ŋ/   /**ŋ**ɜ$^L$/ 'I'                  /l/    /**le**$^H$/ 'earth; ground'
     /ŋ̥/   /**ŋ̥**ɜ$^{ʁH}$/ 'gold'            /l̥/    /**l̥**e$^H$ l̥e$^L$/ 'be thick'

     /r/   /**ro**$^H$/ 'to see: 1sg'
     /ɽ/   /**ɽ**o$^L$/ 'to tear down: 1sg'

4. Rongpa contrasts retroflex fricative /ʐ/ and the trill /r/. Consider these minimal pairs in (4):

(4)  /r/   /**r**ə$^L$/ 'mountain'    /w3$^L$ rə$^H$/ 'bear'
     /ʐ/   /**ʐ**ə$^L$/ 'water'       /w3$^L$ ʐə$^H$/ 'farm cattle'

Phonetically, /r/ can be optionally realized as a retroflex trill, a retroflex flap [ɽ], or even a fricative [ʐ]; /ʐ/, on the other hand, is never pronounced as a trill.

In Rongpa, the orientational prefix *rə-* 'or: upward' can be realized as a glottal stop [ʔə-] or a flap [ɽə-] when combined with verb roots. e.g.,

/rə-ʧʰə$^L$/→[ʔə$^L$-ʧʰə$^H$]~[ɽə$^L$-ʧʰə$^H$] (ort:pfv-to raise (by hand)) 'I have raised upward'
/rə-ʃə$^H$/→[ʔə$^H$-ʃə$^L$]~[ɽə$^H$-ʃə$^L$] (ort:pfv-to put) 'I have put upward'

However, in the interrogative form, only the trill onset [rɛ$^{LH}$-] is produced, it is never realized as [ʔɛ$^{LH}$-]:

/rɛ$^{LH}$-ʧʰə$^L$/→[rɛ$^{LH}$-ʧʰə$^L$] (ort:q-to raise (by hand)) 'Have you raised?'

5. Voiced labiodental fricative /v/ and labial-velar approximant /w/ are contrastive:

(5) /w/ /w3$^L$ z$_{\di:}$$^H$/ 'farm cattle' /w$\varepsilon$$^L$/ 'to bake: 1PL'
/v/ /v3$^H$/ 'tsampa' /k$_{\diː}$$^L$ v$\varepsilon$$^H$/ 'to shake: 1PL'

6. Alveolar nasals /n̪/ and /n/ are realized as a palatal nasal [ɲ̊] and [ɲ], respectively, when followed by the vowels /e, ɛ, o/. In other words, palatal nasals can be observed among the surface forms, but they are not phonemic. Consider the examples below:

(6) /n$\varepsilon$$^L$/→[ɲ$\varepsilon$$^L$] 'fish'
/n̪o$^L$/→[ɲ̊o$^L$] 'fried barley'
/n̪e$^H$ n̪e$^L$/→[ɲ̊e$^H$ ɲ̊e$^L$] 'red'

The alveolar nasal is realized as [n] and [n̪] in other environments.[9] This variation can be formulated as:

$$\begin{bmatrix} +\text{coronal} \\ +\text{nasal} \end{bmatrix} \rightarrow \begin{bmatrix} +\text{dorsal} \\ +\text{high} \\ +\text{front} \end{bmatrix} / \_\_[e, \varepsilon, o]$$

*Alveolar nasal /n, n̪ / is palatalized before [e, ɛ, o].*

7. Velar consonants retract to uvular when the nuclei of the syllable is a uvularized vowel, e.g.,

(7) /ko$^{ʁL}$/→[qo$^{ʁL}$] 'valley'
/k$^h$o$^{ʁH}$/→[q$^h$o$^{ʁH}$] 'head'
/xo$^{ʁL}$/→[χo$^{ʁL}$] 'to wear (necklace): 1SG'
/ɣo$^{ʁL}$/→[ʁo$^{ʁL}$] 'lunatic'

8. The two rather marginal phonemes are: a labio-dental voiceless fricative /f/, only observed in /pa$^{ʁL}$ fe$^H$/ 'soda', and a voiceless velar nasal /ŋ̊/, only observed in /ŋ̊3$^{ʁH}$/ 'gold'.

Table 4 provides example of consonantal phonemes that can serve as a simplex initial:

**Table 4.** Examples of Rongpa simple consonant phonemes.

| Phoneme | Example | Phoneme | Example |
|---------|---------|---------|---------|
| /p/ | /p$_{\diː}$$^H$/ 'sun' | /ʧ/ | /ʧ$_{\diː}$$^L$ ʧ$_{\diː}$$^H$ pa$^{ʁL}$/ 'cicada' |
| /p$^h$/ | /p$^h$$_{\diː}$$^L$/ 'salary' | /ʧ$^h$/ | /ʧ$^h$i$^H$/ 'salt' |
| /b/ | /b$\varepsilon$$^L$ v$_{\diː}$$^H$/ 'pine tree' | /dʒ/ | /dʒ$\varepsilon$$^H$/ 'Han nationality' |
| /m̪/ | /m̪3$^H$/ 'person' | /ʃ/ | /ʃ$_{\diː}$$^H$/ 'incense' |
| /m/ | /mu$^H$/ 'sky' | /ʃ$^h$/ | /ʃ$^h$$_{\diː}$$^H$/ 'worship' |
| /w/ | /w3$^L$ r$_{\diː}$$^H$/ 'bear' | /ʒ/ | /ʒe$^H$/ 'house' |
| /f/ | /pa$^{ʁL}$ fe$^H$/ 'soda' | /tʂ/ | /tʂo$^{ʁH}$/ 'ploughshare' |
| /v/ | /v3$^H$/ 'tsampa' | /tʂ$^h$/ | /tʂ$^h$o$^{ʁH}$/ 'fence' |
| /t/ | /to$^{ʁH}$/ 'wing' | /dʐ/ | /dʐ$_{\diː}$$^H$/ 'to ruminate (animals)' |
| /t$^h$/ | /t$^h$o$^{ʁH}$/ 'meat' | /ʂ/ | /ʂ$_{\diː}$$^H$/ 'louse' |
| /d/ | /do$^L$/ 'to fly' | /ʂ$^h$/ | /ʂ$^h$3$^H$/ 'strength' |
| /ts/ | /ts3$^L$/ 'he' | /ʐ/ | /ʐ$_{\diː}$$^L$/ 'water' |
| /ts$^h$/ | /ts$^h$3$^H$/ 'goat' | /j/ | /$\varepsilon$$^L$ je$^H$/ 'sister-in-law' |
| /dz/ | /dze$^H$/ 'to teach' | /k/ | /ku$^L$/ 'tent' |
| /n̪/ | /ŋ3$^L$ ŋ3$^H$/ 'green; blue' | /k$^h$/ | /k$^h$u$^H$/ 'corner' |

**Table 4.** *Cont.*

| Phoneme | Example | Phoneme | Example |
|---|---|---|---|
| /n/ | /nʒʳᴴ/ 'milk' | /g/ | /goᴴ/ 'supper' |
| /r̥/ | /r̥ɛᴸ/ 'to tear' | /ŋ̊/ | /ŋ̊ʒʳᴴ/ 'gold' |
| /r/ | /rəᴸ/ 'mountain; hill' | /ŋ/ | /ŋɛᴴ/ 'silver' |
| /s/ | /sʒᴴ/ 'pepper' | /x/ | /xəᴴ/ 'mouse' |
| /sʰ/ | /sʰʒᴴ/ 'to kill' | /ɣ/ | /ɣəᴴ/ 'barley liquor' |
| /z/ | /zeᴸ/ 'liver' | | |
| /l̥/ | /l̥aᵏᴴ/ 'wind' | | |
| /l/ | /leᴴ/ 'land; ground' | | |

### 3.2. Consonant Clusters as Complex Initials

Rongpa has a rich inventory of clusters, including (1) clusters with velar fricative /x/ or /ɣ/ as the first element; (2) clusters with nasal /m/ or /N/ as the first element; (3) clusters with bilabial stop /p/ or /b/ as the first element; and (4) clusters with labial glide /w/, as will be discussed in more detail in Sections 3.2.1–3.2.4.

#### 3.2.1. C1 = Dorsal Fricative /x/ or /ɣ/

Table 5 shows all consonant clusters that have dorsal fricatives /x/ or /ɣ/ as the first element C1. C2 in such clusters can be filled by obstruents, voiced nasals, and voiced laterals.

**Table 5.** Inventory of consonant clusters with /x/ or /ɣ/ as the first element C1 (32 clusters).

| | C2 | | Labial | Alveolar | Postalveolar | Retroflex | Velar |
|---|---|---|---|---|---|---|---|
| plosive | voiceless | −ASP | xp | xt | | | xk |
| | | +ASP | xpʰ | xtʰ | | | xkʰ |
| | voiced | | | ɣd | | | ɣg |
| affricate | voiceless | −ASP | | xts | xʧ | xtʂ | |
| | | +ASP | | xtsʰ | xʧʰ | xtʂʰ | |
| | voiced | | | ɣdz | | | |
| nasal | voiced | | xm | xn | | | xŋ |
| | | | ɣm | ɣn | | | ɣŋ |
| fricative | voiceless | −ASP | | xs | xʃ | xʂ | |
| | | +ASP | | xsʰ | xʃʰ | xʂʰ | |
| | voiced | | | ɣz | ɣʒ | ɣʐ | |
| lateral approximant | voiced | | | xl | | | |
| | | | | ɣl | | | |

Some characteristics worth noting regarding these clusters are:

1. The first element /x/ and /ɣ/ are contrastive before voiced nasals and laterals, which is attested by the following minimal pairs in (8):

(8)　/xm/　/xmʒᴴ/ 'crupper-strap'　　/xn/　/xniᴴ/ 'nose'[10]
　　　/ɣm/　/ɣmʒᴴ/ 'mushroom'　　　/ɣn/　/ɣniᴴ/ 'ear'[11]

　　　/xŋ/　/xŋiᴸ/ 'be skewed'　　　/xl/　/xliᴴ/ 'tongue'
　　　/ɣŋ/　/ɣŋiᴴ/ 'drum'　　　　　/ɣl/　/ɣliᴴ/ 'hilt (of knife)'

However, the phonemic contrast between /x/ and /ɣ/ is neutralized when followed by obstruents. That is, /ɣ/ only occurs with voiced obstruents and /x/ only occurs with voiceless ones.

2. In Rongpa, "/x/+voiced sonorants" clusters are not variants of voiceless sonorants. As shown in the following examples in (9), these initials are contrastive.

(9)  /m̥/   /m̥ɜ$^H$/ 'person'          /n̥/   /n̥ɜ$^L$ n̥ɜ$^H$/ 'green; blue'
     /xm/  /xmɜ$^H$/ 'crupper-strap'    /xn/  /xnɜ$^H$ Nbɜ$^L$/ 'mouth'

     /ŋ̊/   /ŋ̊ɜ$^{ʁH}$/ 'gold'         /l̥/   /l̥a$^{ʁH}$/ 'wind'
     /xŋ/  /xŋɜ$^{ʁL}$/ 'parrot'        /xl/  /xla$^{ʁH}$/ 'white poplar'

3. In Rongpa, voiceless aspirated fricatives are phonemic. Note that in Khanggsar (Kǒngsè 孔色) Stau (a Rgyalrongic language spoken in a neighboring area), voiceless fricatives are realized with aspiration in simplex initial position, and as unaspirated when they are a component of a complex initial ([Jacques et al. 2017](), p. 598). However, this is not the case in Rongpa. In this language, voiceless fricatives still contrast in aspiration when preceded by /x/ as C1, as minimal pairs in (10) show:

(10)  /xs/   /xsɜ$^{ʁL}$ xsɜ$^{ʁH}$/ 'be long'        /xʃ/   /xʃə$^{ʁH}$ xʃə$^{ʁH}$/ 'be wide'
      /xs$^h$/  /xs$^h$ɜ$^{ʁL}$ xs$^h$ɜ$^{ʁH}$/ 'be sharp'   /xʃ$^h$/  /xʃ$^h$ə$^{ʁH}$/ 'to break off: 1PL'

      /xʂ/   /xʂɜ$^{ʁL}$ xʂə$^{ʁH}$/ 'to twist'
      /xʂ$^h$/  /xʂ$^h$ə$^H$/ 'to drive away'

4. **The allophones of C1 = /x/ and /ɣ/**

C1 = /x/ and /ɣ/ are observed to surface as uvular and velar fricatives. The distribution of the allophones is laid out as follows:

① Uvular allophones [χ] and [ʁ]

When the nucleus of the syllable is occupied by a uvularized vowel, C1 = /x, ɣ/ are obligatorily retracted to uvular fricative [χ, ʁ] respectively. Plain vowels, however, do not trigger this assimilation. Consider the examples in (11):

(11)  /xpo$^{ʁH}$/→[χpo$^{ʁH}$] 'ice'          /ɣno$^{ʁH}$/→[ʁno$^{ʁH}$] 'to read: 1SG'
      /xpə$^{ʁH}$/→[χpə$^{ʁH}$] 'to blow: 1PL'   /ɣnə$^{ʁH}$/→[ʁnə$^{ʁH}$] 'to read: 1PL'
      /xpe$^{ʁH}$/→[χpe$^{ʁH}$] 'official; leader' /ɣne$^{ʁL}$/→[ʁne$^{ʁL}$] 'to grind'
      /xpɜ$^{ʁH}$/→[χpɜ$^{ʁH}$] 'goiter; pus'    /ɣnɜ$^{ʁH}$/→[ʁnɜ$^{ʁH}$] 'brain'
      /xpa$^{ʁL}$/→[χpa$^{ʁL}$] 'toad'          /ɣna$^{ʁH}$/→[ʁna$^{ʁH}$] 'weed'

② Free variation: [x] or [ʂ]

Voiceless C1 = /x/ can be produced as either [x] or [ʂ]. The two allophones are in free variation. For example:

(12)  /xte$^H$/→[xte$^H$]~[ʂte$^H$] 'mat'
      /xtʂɛ$^H$/→[xtʂɛ$^H$]~[ʂtʂɛ$^H$] 'crupper-strap'
      /xku$^L$ xni$^H$/→[xku$^L$ xni$^H$]~[ʂku$^L$ xni$^H$] 'toe'

Some other free variants can be observed when /x/ is followed by front vowel /i/, which causes /x/ to be fronted to palatal [ç-] or Postalveolar [ʃ]. That is to say, /x/ can surface as either of the four free variants: [x, ʂ, ç, ʃ]. For instance: /xki$^L$/→[ʃki$^L$~çki$^L$~ʂki$^L$~xki$^L$] 'tooth'.

③ Social/contextual variation in /ɣ/:

When produced with an emphatic tone, the voiced C1 = /ɣ/ is sometimes realized as a voiced pharyngeal fricative [ʕ]. In this case, an additional flap [ɾ] or trill-like sound is often produced between [ʕ] and the following consonant, C2.[12] However, this trill/flap is just a concomitant gesture, and cannot be fully produced, otherwise the consultants would say the initial is not pronounced correctly. Consider the examples in (13).

(13)  /ɣni$^H$/→[ʕɾni$^H$] 'ear'
      /ɣnɜ$^H$/→[ʕɾnɜ$^H$] 'tail'

5.  The phonemic status of the first elements /x/ and /ɣ/ is attested by many lexical pairs that are minimally distinguished by the existence or absence of C1:

(14)  /p/    /pa\*ᴸ/ 'be deaf'                          /pʰ/    /pʰoᵏᴴ/ 'to splash: 1sɢ'
      /xp/   /xpa\*ᴸ/ 'toad'                            /xpʰ/   /xpʰoᵏᴸ/ 'to cover: 1sɢ'

      /m/    /meᴴ/ 'to name'                            /t/     /toᵏᴴ/ 'wing'
      /xm/   /xmeᴴ/ 'medicine'                          /xt/    /xtoᵏᴴ/ 'sink; Tibetan silver'

      /x/    /tʰəᴴ/ 'cereal (barley and wheat)'         /d/     /dəᵏᴸ/ 'to saw: 1ᴘʟ'
      /xt/   /xtʰəᴴ/ 'to stampede'                      /ɣd/    /ɣdəᵏᴴ/ 'umbrella'

      /ts/   /tsəᵏᴴ/ 'grate'                            /tsʰ/   /tsʰaᵏᴴ paᵏᴸ/ 'wink; blink'
      /xts/  /xtsəᵏᴴ/ 'ecphyma'                         /xtsʰ/  /xtsʰaᵏᴴ/ 'cough [N]'

      /dz/   /dzəᵏᴸ/ 'to thread: 1ᴘʟ'                    /z/     /zoᵏᴴ/ 'woman; daughter'
      /ɣdz/  /ɣdzəᵏᴴ/ 'pillar'                          /ɣz/    /ɣzoᵏᴸ/ 'storage container'

      /s/    /s₃ᵏᴸ piᴴ/ 'be new'                        /n/     /naᵏᴸ laᵏᴸ/ 'leaf'
      /xs/   /xs₃ᵏᴸ xs₃ᵏᴴ/ 'be long'                    /xn/    /xnaᵏᴸ/ 'nettle'

      /l/    /kʰeᴴ ləᴴ/ 'spleen'                        /ʧ/     /ʧɛᴸ bʒɛᴴ/ 'clay teapot'
      /xl/   /xləᴸ/ 'birch'                             /xʧ/    /xʧɛᴸ bʒɛᴴ/ 'shovel'

      /ʧʰ/   /ʧʰəᵏᴸ ʧʰəᵏᴴ/ 'be sweet'                   /tʂ/    /tʂuᴸ/ 'village'
      /xʧʰ/  /xʧʰəᵏᴴ/ 'dog'                             /xtʂ/   /xtʂuᴴ məᴴ/ 'beggar'

      /tʂʰ/  /tʂʰəᵏᴸ tʂʰəᵏᴴ/ 'be pleasant'             /ʂ/     /ʂoᵏᴸ ʂaᵏᴴ/ 'peach tree'
      /xtʂʰ/ /xtʂʰəᵏᴴ/ 'to carry (in arms): 1ᴘʟ'       /xʂ/    /xʂaᵏᴸ/ 'soil'

      /ʂʰ/   /ʂʰ₃ᴴ/ 'strength'                         /ʐ/     /ʐ₃ᴸ/ 'travel-ready dishes'
      /xʂʰ/  /xʂʰ₃ᴴ/ 'to drive (cattle): 1sɢ'          /ɣʐ/    /ɣʐ₃ᴴ/ 'fertilizer (excrements)'

      /k/    /koᴸ/ 'porcupine'                          /kʰ/    /kʰuᴴ/ 'corner'
      /xk/   /xkoᴸ/ 'to carry (on shoulder):1sɢ'        /xkʰ/   /xkʰuᴴ/ 'smoke'

      /g/    /geᴸ/ 'to like: 2ᴘʟ'
      /ɣg/   /moᴴ ɣgeᴸ/ 'old woman'

Table 6 lists all the examples of complex initials with /x, ɣ/ as C1:

**Table 6.** Examples of consonant clusters with /x, ɣ/ as a first element.

| Cluster | Example | Cluster | Example |
|---------|---------|---------|---------|
| /xp/ | /xpeᴸ/ 'urine' | /xʧ/ | /xʧeᴴ/ 'warehouse' |
| /xpʰ/ | /xpʰoᵏᴸ/ 'to bury' | /xʧʰ/ | /xʧʰəᵏᴴ/ 'dog' |
| /xm/ | /xm₃ᴴ/ 'crupper-strap' | /xʃ/ | /xʃəᵏᴸ xʃəᵏᴴ/ 'be wide' |
| /ɣm/ | /ɣmaᵏᴴ/ 'bamboo' | /xʃʰ/ | /xʃʰ₃ᵏᴴ/ 'to scatter (with hands)' |
| /xt/ | /xtaᵏᴸ/ 'wall' | /ɣʒ/ | /ɣʒəᵏᴸ/ 'to blend' |
| /xtʰ/ | /xtʰəᴸ/ 'to stamp' | /xtʂ/ | /xtʂeᴴ/ 'cloud' |
| /ɣd/ | /ɣdəᵏᴴ/ 'umbrella' | /xtʂʰ/ | /xtʂʰaᵏᴴ/ 'sifter' |
| /xts/ | /xtsəᴸ/ 'hail' | /xʂ/ | /xʂaᵏᴸ/ 'soil' |
| /xtsʰ/ | /xtsʰaᵏᴴ/ 'cough' | /xʂʰ/ | /xʂʰ₃ᴸ/ 'to chase (cattle)' |
| /ɣdz/ | /ɣdz₃ᵏᴴ/ 'pillar' | /ɣʐ/ | /ɣʐ₃ᴴ/ 'fertilizer' |
| /xn/ | /xnaᵏᴸ/ 'to peel' | /xk/ | /xk₃ᴴ/ 'sound' |

**Table 6.** *Cont*.

| Cluster | Example | Cluster | Example |
|---|---|---|---|
| /ɣn/ | /ɣn₃$^H$/ 'tail' | /xkʰ/ | /xkʰu$^H$/ 'smoke' |
| /xs/ | /xso$^{ʁL}$/ 'pine oil' | /ɣg/ | /xku$^H$ ɣge$^H$/ 'thief' |
| /xsʰ/ | /xsʰ₃$^{ʁL}$ xsʰ₃$^{ʁH}$/ 'be sharp' | /xŋ/ | /xŋ₃$^{ʁL}$/ 'parrot' |
| /ɣz/ | /ɣzə$^{ʁH}$/ 'agate' | /ɣŋ/ | /ɣŋi$^H$/ 'drum' |
| /xl/ | /xla$^{ʁH}$/ 'white poplar' | | |
| /ɣl/ | /ɣlə$^H$/ 'hand' | | |

### 3.2.2. C1 = Nasal /m/ or /N/

Rongpa has 21 complex initials with a nasal as the first element. C2 that are compatible with /m/ and /N/ are stops and affricates, either voiced or voiceless aspirated. Note that C2 cannot be a voiceless unaspirated stop or affricate. Additionally, /m/ can also occur before a voiced nasal /n/.

Table 7 below exhibits the inventory of consonant clusters with a nasal as the first element:

**Table 7.** Inventory of consonant clusters with /m/ or /N/ as a first element. (21 clusters).

| C2 | | | Labial | Alveolar | Postalveolar | Retroflex | Velar |
|---|---|---|---|---|---|---|---|
| plosive | voiceless | +ASP | | mtʰ | | | |
| | | | Npʰ | Ntʰ | | | Nkʰ |
| | voiced | −ASP | | md | | | |
| | | | Nb | Nd | | | Ng |
| affricate | voiceless | +ASP | | mtsʰ | mʧʰ | mtʂʰ | |
| | | | Ntsʰ | | Nʧʰ | Ntʂʰ | |
| | voiced | −ASP | | mdz | mdʒ | mdʐ | |
| | | | Ndz | | Ndʒ | Ndʐ | |
| nasal | voiced | | | mn | | | |

Some characteristics worth noting regarding these clusters are:

1. The abstract phoneme /N-/ is homorganic to the following consonant, which can be stated using this phonological rule:[13]

$$N- \rightarrow [\text{place}_i]/\underline{\quad} \begin{bmatrix} +\text{consonantal} \\ \text{place}_i \end{bmatrix}$$

*N- assimilates to the place of articulation of the following consonant.*

For example:

(15)  /Npʰa$^{ʁL}$ ra$^{ʁH}$/→[mpʰa$^{ʁL}$ ra$^{ʁH}$] 'to scratch'
/Ntʰe$^H$/→[ntʰe$^H$] 'to sing: 1SG'
/Nkʰu$^L$/→[ŋkʰu$^L$] 'to use'

2. C1 = /m/ and /N/ are contrastive before voiced plosives or affricates, as demonstrated by the examples in (16):

(16)  /md/   /mda$^{ʁH}$/ 'bow'        /mdz/  /mdz₃$^H$/ 'room'
/Nd/   /a$^{ʁL}$ Nda$^{ʁH}$/ 'bull'     /Ndz/  /Ndz₃$^H$/ 'to insert'

/mdʒ/  /mdʒ₃$^L$/ 'be sticky'   /mdʐ/  /mdʐɛ$^H$/ 'rice'
/Ndʒ/  /Ndʒ₃$^L$/ 'to change'   /Ndʐ/  /Ndʐɛ$^L$ bə$^H$/ 'fruit'

3. If it is preceded by /m/ and the nucleus is /ɛ/ or /a$^ʁ$/, C2 can be labialized (adding the features of [+labial] and [+round]). Related examples are:

(17)  /mtʂʰa$^{ʁH}$/→[mtʂʰʷa$^{ʁH}$] 'plant ash'
/mdʐɛ$^H$/→[mdʐʷɛ$^H$] 'rice'

This alternation can be formulated by this phonological rule:

$$\begin{bmatrix} -\text{syllabic} \\ +\text{consonantal} \end{bmatrix} \rightarrow \begin{bmatrix} +\text{labial} \\ +\text{round} \end{bmatrix} / \ m \underline{\quad} \begin{bmatrix} +\text{syllabic} \\ +\text{low} \end{bmatrix}$$

*The second consonant is labialized when preceding a /m-/ and followed by a low vowel.*

This alternation also gives evidence that [mpʰ] should be underlyingly analyzed as /Npʰ/ rather than /mpʰ/, since in surface realization, no labialization is produced:

/Npʰaᵂᴸ raᵂᴴ/→[**m**pʰaᵂᴸ raᵂᴴ] *[**mpʰʷ**aᵂᴸ raᵂᴴ]

In surface realization, some instances of [m] in C1 position are actually derived from the coalescence of /N/ and a preceding third-person prefix *P-*,[14] as demonstrated in example (18).

(18) /**P-N**tʰeᴴ/→[**m**tʰeᴴ] 'he sings'
/**P-N**dʐəᵂᴸ/→[**m**dʐəᵂᴸ] 'he arrests'

4. The 'nasal + stop/affricate' clusters can also be found across syllable boundaries. Syllable boundary readjustment has been applied, with original coda becoming part of the following onset. Many of the related examples are etymologically Tibetan loanwords. Examples are listed in Table 8:

**Table 8.** Marginal "nasal+stop/affricate" clusters in word medial position.

| Rongpa | WT | Gloss |
|---|---|---|
| [ndaᵂᴸ.**mb**aᵂᴴ] | འདམ་པ <'dam.pa> | 'mud' |
| [ʃuᴴ.**ŋkʰ**uᴴ] | སྤྱང་ཀི <spyang.ki> | 'wolf' |
| [loŋᴸ.**ŋkʰ**əᴴ] | རླུང <rlung> | 'blower (used to chaff)' |
| [bdʐaᵂᴸ.**mb**aᵂᴴ] | འགྲམ་པ <'gram.pa> | 'cheek' |
| [loŋᴸ.**tʂʰ**əᴴ][15] | རླངས <rlangs> | 'food steamer' |
| [toŋᴸ.**tʂʰ**aᵂᴴ.təᴸ.roᴸ] | སྟོང <stong> | '(one) thousand' |

5. The phonemic status of the first elements /N/ and /m/ is attested by these minimal pairs in (19):

(19)

| | | | |
|---|---|---|---|
| /pʰ/ | /**pʰ**aᵂᴴ/ 'to vomit' | /b/ | /**b**uᴴ/ 'breath' |
| /Npʰ/ | /**Npʰ**aᵂᴸ raᵂᴴ/ 'to scratch' | /Nb/ | /**Nb**oᴴ/ 'to protect:1sɢ' |
| /tʰ/ | /**tʰ**iᴴ/ 'big water tank' | /d/ | /**d**eᴴ/ 'to accumulate: 2ᴘʟ' |
| /mtʰ/ | /**mtʰ**iᴸ **mtʰ**iᴴ/ 'be tall' | /Nd/ | /**Nd**eᴴ/ 'what' |
| /tsʰ/ | /gəᴸ **tsʰ**iᴴ/ 'vertebra' | /dz/ | /**dz**ɛᴴ/ 'to teach: 1ᴘʟ' |
| /mtsʰ/ | /**mtsʰ**iᴴ/ 'lake' | /mdz/ | /**mdz**ɛᴸ/ 'to soot' |
| /dz/ | /**dz**əᵂᴴ/ 'to thread: 1ᴘʟ' | /n/ | /**n**əᴸ/ 'forest' |
| /Ndz/ | /**Ndz**əᵂᴴ/ 'a unit of length'[16] | /mn/ | /**mn**əᴴ/ 'eye' |
| /ʧʰ/ | /**ʧʰ**ɜᴴ **ʧʰ**ɜᴸ/ 'be thin' | /ʤ/ | /**ʤ**oᴴ/ 'to run: 1sɢ' |
| /mʧʰ/ | /**mʧʰ**ɜᴸ **mʧʰ**ɜᴴ/ 'be pretty' | /Nʤ/ | /**Nʤ**oᴸ **Nʤ**oᴴ/ 'be weak (liquor)' |
| /tʂʰ/ | /**tʂʰ**ɜᴸ/ 'to cut' | /dʐ/ | /**dʐ**ɛᴴ/ 'enemy' |
| /mtʂʰ/ | /**mtʂʰ**ɜᴴ **mtʂʰ**ɜᴸ/ 'be skewed' | /mdʐ/ | /**mdʐ**ɛᴴ/ 'rice' |
| /kʰ/ | /**kʰ**uᴴ/ 'corner' | /g/ | /**g**uᴸ/ 'be freezing' |
| /Nkʰ/ | /**Nkʰ**uᴸ/ 'to use: 3' | /Ng/ | /**Ng**uᴴ **Ng**uᴸ/ 'to nod (head)' |

Table 9 lists all the example of consonant clusters with /m/ or /N/ as the first element:

**Table 9.** Examples of consonant clusters with /m/ or /N/ as a first element.

| Cluster | Example | Cluster | Example |
|---|---|---|---|
| | | /Npʰ/ | /**Npʰ**aˠL raˠH/ 'to scratch' |
| | | /Nb/ | /**Nb**eH/ 'to protect' |
| /mtʰ/ | /**mtʰ**iL **mtʰ**iH/ 'be tall' | /Ntʰ/ | /**Ntʰ**eH/ 'to sing' |
| /md/ | /**md**iH/ 'meal' | /Nd/ | /**Nd**eH/ 'what' |
| /mtsʰ/ | /**mtsʰ**iH/ 'lake' | /Ntsʰ/ | /**Ntsʰ**oL/ 'nest' |
| /mdz/ | /**mdz**ɝH/ 'room' | /Ndz/ | /**Ndz**ɝH/ 'to insert' |
| /mtʃʰ/ | /**mtʃʰ**ɝˠL **mtʃʰ**ɝˠH/ 'be pretty' | /Ntʃʰ/ | /tʂiL **Ntʃʰ**uH/ 'horse bell' |
| /mdʒ/ | /**mdʒ**ɝH/ 'be sticky' | /Ndʒ/ | /**Ndʒ**əL/ 'trouser' |
| /mtʂʰ/ | /**mtʂʰ**aˠH/ 'plant ash' | /Ntʂʰ/ | /**Ntʂʰ**eH/ 'be expensive' |
| /mdʐ/ | /**mdʐ**iH/ 'thunder' | /Ndʐ/ | /**Ndʐ**eL/ 'selvedge' |
| /mn/ | /**mn**əH/ 'eye' | /Nkʰ/ | /**Nkʰ**uL/ 'to use: 3' |
| | | /Ng/ | /**Ng**uH **Ng**uL/ 'to nod' |

### 3.2.3. C1 = Bilabial Fricative /p/ or /b/

Table 10 presents all complex initials with a bilabial stop as the first element. C2 that are compatible with bilabial C1 are basically coronal consonants, except for /bv/, in which labiodental voiced fricative /v/ serves as C2.

**Table 10.** Inventory of consonant clusters with /p/ or /b/ as the first element. (27 clusters).

| C2 | | | Labial | Alveolar | Postalveolar | Retroflex |
|---|---|---|---|---|---|---|
| plosive | voiceless | −ASP | | pt | | |
| | | +ASP | | ptʰ | | |
| | voiced | | | (bd) | | |
| affricate | voiceless | −ASP | | pts | ptʃ | ptʂ |
| | | +ASP | | ptsʰ | ptʃʰ | ptʂʰ |
| | voiced | | | (bdz) | (bdʒ) | (bdʐ) |
| trill | voiceless | | | pr̥ | | |
| | voiced | | | br | | |
| fricative | voiced | −ASP | | ps | pʃ | pʂ |
| | | +ASP | | psʰ | pʃʰ | pʂʰ |
| | voiced | | bv | bz | bʒ | bʐ |
| lateral approximant | voiceless | | | pl̥ | | |
| | voiced | | | pl | | |
| | | | | bl | | |

Some characteristics worth noting regarding these clusters are:

1. The phonemic status of the first element /p/ and /b/

Among the clusters in Table 10, the voicing contrast between /p/ and /b/ is only attested before lateral /l/, as the following minimal pairs in (20) demonstrate:[17]

(20) /pl/ /**pl**oL/→[**ɸl**oL] 'to plait: 1sɢ' /kʰoˠL **pl**əH/→[qʰoˠL **ɸl**əH] 'to unbraid hair'
     /bl/ /**bl**oH/→[**βl**oH] 'to do: 1sɢ' /koˠL **bl**əH/→[qoˠL **βl**əH] 'lightning'

However, the contrast between /p/ and /b/ is neutralized before obstruents and trills; we find only [p]~[ɸ] before voiceless obstruents and voiceless trills, and [b]~[β] only before their voiced counterparts.

2.  Allophones of the first element /p/ and /b/

The first element C1 shows a process of manner assimilation, specifically: when the second element C2 is a stop or affricate, C1 remain a bilabial stop [p] or [b] (see examples in (21)); however, when C2 is a fricative or liquid, C1 is realized as a bilabial fricative [ɸ] or [β], as examples in (22) show:

(21)  /ptʂoʁH/→[**pt**ʂoʁH] 'bull'
/ptʂhəʁH/→[**pt**ʂhəʁH] 'tusk (of animals)'
/ptɕəH ptɕəL/→[**pt**ɕəH **pt**ɕəL] 'temple '
/bdʐ3H→[**bd**ʐ3H] 'be full'
/bdʐuL /→[**bd**ʐuL] 'to extend'

(22)  /pʃaʁH/→[**ɸ**ʃaʁH] 'honey'
/psəH/→[**ɸ**səH] 'light'
/bʒeL/→[**β**ʒeL] 'scabbard'
/blaʁH/→[**β**laʁH] 'thigh'
/braʁH/→[**β**raʁH] 'cow'

The manner assimilation rule can be formulated as:

$$\begin{bmatrix} p- \\ b- \end{bmatrix} \rightarrow [\alpha \text{ continuant}] / \underline{\quad} \begin{bmatrix} +\text{consonantal} \\ \alpha \text{ continuant} \end{bmatrix}$$

*/p/ and /b/ take on the manner of articulation (regarding the feature [±continuant]) of the following consonant.*

It should be noted that, although at first glance the surface representation of [b] as the first element can be accounted for using the manner-assimilation rule above, most of the related examples are Tibetan loanwords, so the complex initials with [b] could have been imported directly from neighboring Tibetan dialects.[18] Consider the examples in Table 11.

**Table 11.** Marginal clusters of [bC] in Rongpa (question mark "?"denotes uncertainty of the etymological origins).

| Cluster | Example | WT | Gloss |
|---|---|---|---|
| [bd] | [niL xuH tsaʁL **bd**əH] | བདུན <**bd**un> | 'twenty-seven' |
| [bdz] | [shaʁL **bdz**3H muL təL] | (ཨཛི <**brdz**i>)? | 'cosmos flower' |
| [bdʐ] | [**bdʐ**aʁL mbaʁH] | འགྲམ་པ <'**gram**.pa> | 'cheek' |
| [bdʒ] | [**bdʒ**3H] | (རྒྱག <**rgy**ag>)? | 'be full' |
| | [niL xuH tsaʁL **bdʒ**eH] | བརྒྱད <**brgy**ad> | 'twenty-eight' |
| | [**bdʒ**uL] | (བརྒྱངས <**brgy**angs>)? | 'to extend' |

3.  Analogy with /m/ discussed above; the first element /p/ also triggers labialization of the following consonant:

(23)  /**p**tsaʁH/→[ptsʷaʁH] 'rust'[19]
/**p**tʂεL/→[ptʂʷεL] 'distiller's yeast; friend; placenta'
/**p**ʃaʁH/→[ɸʃʷaʁH] 'honey'
/dʒεL **b**ʒεH/→[dʒεL βʒʷεH] 'chicken'

Thus, the phonological rule stated above can be further expanded as:

$$\begin{bmatrix} -\text{syllabic} \\ +\text{consonantal} \end{bmatrix} \rightarrow \begin{bmatrix} +\text{labial} \\ +\text{round} \end{bmatrix} / \begin{bmatrix} +\text{labial} \\ +\text{round} \end{bmatrix} \underline{\quad} \begin{bmatrix} +\text{syllabic} \\ +\text{low} \end{bmatrix}$$

*The second consonant is labialized when preceding a labial consonant and followed by a low vowel.*

4.　Again, the contrast of fricative aspiration still exists in /pC/ clusters:

(24)　/ps/　/**ps**o^H/ 'to grind: 1SG'　　　　　/pʃ/　/**pʃ**aʶ^H/ 'honey'
　　　/psʰ/　/**psʰ**o^H/ 'to select: 1SG'　　　　/pʃʰ/　/**pʃʰ**aʶ^H/ 'cedar'

　　　/pʂ/　/**pʂ**ə^H/ 'to speak: 1PL'
　　　/pʂʰ/　/**pʂʰ**ə^H/ 'to go: 3 (light verb)'

5.　Apart from prefix *P-* in inflected verbs, there are also nominal examples showing that the status of the bilabial C1 is stable in Rongpa phonology; see minimal pairs in (25):

(25)　/ts/　/**ts**aʶ^H/ 'fat (meat)'　　　　　/r/　/**r**e^H/ 'cliff'
　　　/pts/　/**pts**aʶ^H/ 'rust'　　　　　　/br/　/**br**e^H/ 'horse'

　　　/s/　/**s**i^L **s**i^H/ 'scarecrow'　　　/l/　/**l**e^H/ 'ground; earth'
　　　/ps/　/**ps**i^H/ 'moth (in wool)'　　　/pl/　/**pl**e^L/ 'wooden plate'

　　　/tʃ/　/**tʃ**ə^L **tʃ**ə^H paʶ^L/ 'cicada'　　　/ʒ/　/**ʒ**e^H/ 'sheep'
　　　/ptʃ/　/**ptʃ**ə^H **ptʃ**ə^L/ 'temple (of the body)'　　/bʒ/　/**bʒ**e^L/ 'scabbard'

　　　/tʂ/　/**tʂ**oʶ^L/ 'ploughshare'
　　　/ptʂ/　/**ptʂ**oʶ^H/ 'bull'

6.　The combination of /bv/ is rather marginal since it is observed only once in our database, namely /bve^H/→[βve^H] 'pig'. The transcription of 'pig' as /bve^H/→[βve^H] is also attested by the 3^rd person form of /kə^L ve^H/ 'to shake', that is, /kə^L **P**-ve^H/→[kə^L βve^H] 'he shakes'. My consultant judged the second syllable [βve^H] as homophonic with /bve^H/→[βve^H] 'pig'.

7.　Note that except for /bv/, C2 = labial and velar consonants are almost absent in Table 9. It appears that clusters such as [ppʰ], [pk], and [bg] are not allowed for the phonological restrictions in this language. This fact can also be attested by the morphological evidence of the third-person prefix *P-* attached to the verb root with bilabial and velar consonant initials. See examples in (26):

(26)　/**P**-pe^H/→[pe^H] 'He digs'
　　　/**P**-kʰe^L/→[kʰe^L] 'He airs (the clothes)'

　At first glance, it seems that both /p/ and /kʰ/ are immune to being prefixed with *P-*. However, this third-person prefix *P-* can be realized as a glide [w] between velar consonant initials and an uvularized open /aʶ/. See examples in (27):

(27)　/**P**-kʰaʶ^L/→[qʰ**w**aʶ^L] 'He picks'
　　　/**P**-xkaʶ^H/→[χq**w**aʶ^H] 'He carves'

　It is more plausible to state that the third-person prefix *P-* is not truly absent. It is realized as a glide [w], and this glide can be deleted in specific environments. Specifically, take the verb /kʰe^L/ 'to air' as an example: its underlying third-person form is /kʰwe^L/ and its surface form is [kʰe^L] due to the /w/ deletion. Alternatively, we can also state that /e/ and /we/ are neutralized after /kʰ/. More detailed information is pending future research.
　Examples of consonant clusters with /p/ or /b/ as the first element are listed in Table 12:

**Table 12.** Examples of consonant clusters with /p/ or /b/ as a first element.

| Cluster | Example | Cluster | Example |
|---|---|---|---|
| /bv/ | /**bv**e^H/ 'pig' | /pʧ/ | /**pʧ**ə^H **pʧ**ə^L/ 'temple (of the body)' |
| /pt/ | /**pt**aʁ^H/ 'to twine: 3' | /pʧʰ/ | /**pʧʰ**i^H/ 'to eat: 3' |
| /ptʰ/ | /**ptʰ**e^H/ 'to drink: 3' | /bʤ/ | /**bʤ**ɜ^L/ 'be full' |
| /bd/ | /ni^L xu^H tsaʁ^L **bd**ə^H/ 'twenty-seven' | /pʃ/ | /**pʃ**aʁ^H/ 'honey' |
| /pts/ | /**pts**e^H/ 'ghost' | /pʃʰ/ | /**pʃʰ**aʁ^H/ 'cedar' |
| /ptsʰ/ | /**ptsʰ**ɛ^H/ 'to dye: 3' | /bʒ/ | /**bʒ**e^L/ 'sheath' |
| /bdz/ | /sʰaʁ^L **bdz**ɜ^H mu^L tə^L/ 'cosmos flower' | /ptʂ/ | /**ptʂ**ɛ^L/ 'friend' |
| /pɻ̥/ | /**pɻ̥**ɛ^L/ 'to tear down: 3' | /ptʂʰ/ | /**ptʂʰ**ə^H ro^H/ 'six' |
| /br/ | /**br**aʁ^H/ 'cow' | /bdʐ/ | /**bdʐ**aʁ^L Nbaʁ^H/ 'cheek' |
| /ps/ | /**ps**i^H/ 'worm' | /pʂ/ | /**pʂ**ə^H/ 'to speak: 3' |
| /psʰ/ | /**psʰ**ɜ^H/ 'to kill: 3' | /pʂʰ/ | /**pʂʰ**ə^H/ 'to go (light verb): 3' |
| /bz/ | /**bz**i^H/ 'marmot' | /bʐ/ | /**bʐ**ə^H ro^L/ 'four' |
| /pl̥/ | /**pl̥**oʁ^H/ 'to knead: 3' | | |
| /pl/ | /**pl**e^H/ 'wooden plate' | | |
| /bl/ | /**bl**aʁ^H/ 'thigh' | | |

### 3.2.4. Clusters with Glide /w/

As described in Section 2, the occurrence of /w/ in consonant clusters requires a restricted environment; /w/ only occurs after velar obstruents, and the nuclei must be [+low] vowels, specifically /ɛ, aʁ/. There are only 4 clusters with glide /w/ in Rongpa, which are listed in Table 13:

**Table 13.** Examples of consonant clusters with glide /w/.

| Cluster | Example | Cluster | Example |
|---|---|---|---|
| /kw/ | /**kw**aʁ^H loʁ^L/ 'iron stove' | /xkw/ | /**xkw**ɛ^H/ 'be shy' |
| /kʰw/ | /**kʰw**aʁ^L/ 'to pick: 3' | | |
| /gw/ | /**gw**ɛ^L/ 'to block: 3' | | |

Minimal pairs that show the contrastive status of glide /w/ are:

(28)  /**k**aʁ^L/ 'drop down'      /**kʰ**aʁ^L **kʰ**aʁ^H/ 'be angry'
      /**kw**aʁ^H loʁ^L/ 'iron stove'   /**kʰw**aʁ^L/ 'to pick: 3'

      /**g**ɛ^L/ 'to owe: 1PL'      /**xk**ɛ^L/ 'to carry (on shoulder): 1PL'
      /**gw**ɛ^L/ 'to block: 3'      /**xkw**ɛ^L/ 'charcoal'

As briefly described in Footnote 14, glide [w] also occurs as an irregular allomorph of third-person prefix *P-*. When the initial of the verb root includes a velar stop and the vowel of the root is uvularized /aʁ/, a glide [w] is inserted. For example:

(29)  /**P**-kʰaʁ^L/→[qʰ**w**aʁ^L] 'He picks'
      /**P**-xkaʁ^H/→[xq**w**aʁ^H] 'He carves'

That is to say, the glide [w] could be a surface realization of the 3rd person prefix *P-*. It follows that at least some [kʰw] and [xkw] clusters can be analyzed underlyingly as /P-kʰ/ or /P-xk/ respectively. More detailed analysis of this morphophonemic process is in order in future research.

## 4. Rhymes

As shown in Table 14, vowels in Rongpa can be divided into two types: plain and uvularized ones, which make up thirteen vocalic phonemes in total. Sections 4.1 and 4.2 present a phonemic analysis of the vowels, and the phenomena of vowel harmony will be discussed in Section 4.3.

**Table 14.** Rongpa vowel phonemes inventory.

| Plain Vowels | i | e | ɛ | ɘ | ə | ɜ | o | u |
|---|---|---|---|---|---|---|---|---|
| Uvularized vowels | | eʁ | aʁ | | əʁ | ɜʁ | oʁ | |

The following examples in Table 15 illustrate the vocalic phonemes:

**Table 15.** Examples of Rongpa vowel phonemes.

| Phoneme | Examples | |
|---|---|---|
| /i/ | /zi$^L$/ 'wool' | /ʧʰi$^H$/ 'to eat' |
| /e/ | /le$^H$/ 'ground; earth' | /tʰe$^H$/ 'to drink' |
| /ɛ/ | /rɛ$^L$/ 'cloth' | /psʰɛ$^H$/ 'to select $_{(seed)}$' |
| /ɘ/ | /nɘ$^L$/ 'forest' | /ʒɘ$^H$/ 'to take way; to move' |
| /ə/ | /pə$^H$/ 'sun' | /ʧʰə$^H$/ 'to lift; to raise' |
| /ɜ/ | /pɜ$^H$/ 'photo' | /tsɜ$^H$/ 'to milk $_{(cow)}$' |
| /o/ | /ko$^L$/ 'porcupine' | /xko$^H$/ 'to believe' |
| /u/ | /mu$^H$/ 'sky' | /xku$^L$/ 'to give birth $_{(a\ child)}$' |
| /oʁ/ | /zoʁ$^H$/ 'woman' | /ʐoʁ$^L$/ 'to promise' |
| /əʁ/ | /sʰəʁ$^H$/ 'blood' | /Ndʐəʁ$^L$/ 'to support $_{(with\ hands)}$' |
| /eʁ/ | /xpeʁ$^H$/ 'official; leader' | /xʃʰeʁ$^H$/ 'to snap $_{(with\ hands)}$: 2' |
| /ɜʁ/ | /mɜʁ$^L$/ 'fire' | /xnɜʁ$^H$/ 'to shine; to light up' |
| /aʁ/ | /ɣaʁ$^H$/ 'door' | /xkaʁ$^H$/ 'to carve' |

### 4.1. Plain Vowels

Rongpa distinguishes eight plain vowel phonemes, see Figure 2:

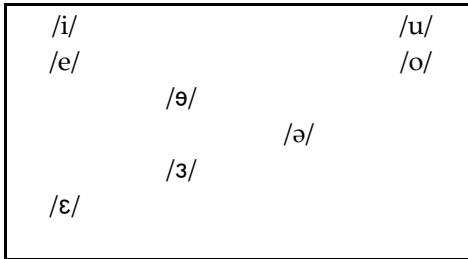

**Figure 2.** Rongpa plain vowels inventory.

Phonemic analysis of plain vowels:

1. Minimal sets of all eight plain vowels

As shown in (30), all the plain vowels are contrastive after voiced velar stop /g/:

(30) /i/ /gi$^H$/ 'to like:1PL'

/e/ /ge$^H$/ 'to ripe (fruit)'

/ɛ/ /gɛ$^H$/ 'saddle'

/ɘ/ /gɘ$^H$/ 'vulture'

/ɜ/ /gɜ$^H$/ 'to warm oneself (by a fire): 1SG'

/ə/ /gə$^H$/ 'to warm oneself (by a fire): 1PL'

/o/ /go$^H$/ 'supper'

/u/ /gu$^L$/ 'be freezing; to owe (money)'

2. Closed vowels /i/ and /u/

The phonemes /i/ and /u/ are high but slightly centralized vowels, which should be phonetically transcribed as [ɨ] and [ʉ], respectively. Regarding the distribution of [ɨ] and [ʉ], as Table 16 shows, [ʉ] never occurs after coronal onsets. [ɨ], on the other hand, occurs more frequently after coronal onsets. However, the two vowels are not in complementary distribution, as they are contrastive after labial and velar consonants:

**Table 16.** The distribution of [ɨ] and [ʉ] (i.e., the places of the preceding consonants).

|  | **Labial** | **Alveolar** | **Postalveolar** | **Retroflex** | **Velar** |
|---|---|---|---|---|---|
| [ɨ] | (√)[20] | √ | √ | √ | √ |
| [ʉ] | √ | – | – | – | √ |

Minimal pairs that show the contrast between /i/ and /u/ are in (31):

(31) /i/ /mi$^H$/ 'wound' /gi$^H$/ 'to like: 1PL'

/u/ /mu$^H$/ 'sky' /gu$^L$/ 'to owe (money)'

3. Close-mid central vowel /ɘ/

/ɘ/ is a rather illusive phoneme. Minimal pairs that show /ɘ/ is distinct from other adjacent vowels are shown in (32):

(32) /ɘ/ /xpɘ$^H$/ 'to invert' /ʒe$^H$/ 'sheep'

/e/ /xpe$^H$/ 'Tibetan incense' /ʒɘ$^H$/ 'to move'

/ə/ /xpə$^H$/ 'to soak: 1PL' /ʒə$^H$/ 'to sleep: 1PL'

When /ɘ/ occurs after labial and velar consonants, an insertion of glide /w/ is applied, as in the examples in (33), while after coronal consonants, this glide /w/ is absent. See examples in (34):

(33) /pɘ$^L$/→[pwɘ$^H$] 'cow dung'

/mɘ$^H$/→[mwɘ$^H$] 'butter'

/xɘ$^H$/→[xwɘ$^H$] 'mouse'

/gɘ$^H$/→[gwɘ$^H$] 'vulture'

(34) /ɣlɘ$^H$/→[ɣlɘ$^H$] 'hand'

/xtɘ$^H$/→[xtɘ$^H$] 'excrement'

/xnɘ$^H$/→[xnɘ$^H$] 'sisters (of a man)'

/t$^h$ɘ$^H$/→[t$^h$ɘ$^H$] 'grain (barley and wheat)'

This skewed distribution in which the inserted [w] occurs only after grave consonants (i.e., labial and dorsal) reminds us of the conspicuous phonotactic constrain mentioned in Section 3.2.3. That is, labial stops [p-] and [b-] occur only before coronal consonants. These evidences prompt some hypothesis. For one thing, clusters [pw], [mw], [xw], and [gw] in (33) might be other irregular realizations of labial and velar consonants that prefixed with [p-] or [b-] (also see [w] as an irregular allomorph of 3$^{rd}$ person prefix *P-* in endnote 14). For another thing, the independent vowel phoneme /ɘ/ in (34) emerges through a vowel coalescence with glide [w], specifically, *wV→/ɘ/.

4. Close-mid front vowels /e/ and back /o/

/e/ and /o/ are realized as [ɪ] and [ʊ] phonetically. Note that these two vowels are distinguishable from /i/→[i] and /u/→[ʉ], respectively, in this language, as shown in the minimal pairs in (35) and (36):

(35) /i/ /ʒiʰ/→[ʒiʰ] 'face'  /ziᴸ/→[ziᴸ] 'wool'
     /e/ /ʒeʰ/→[ʒɪʰ] 'sheep'  /zeᴸ/→[zɪᴸ] 'liver'

(36) /u/ /kuᴸ/→[kʉᴸ] 'tent'  /xkuᴸ/→[xkʊᴸ] 'to give birth ₍child₎'
     /o/ /koᴸ/→[kʊᴸ] 'porcupine'  /xkoᴸ/→[xkʉᴸ] 'to carry ₍on shoulder₎'

5.   Central vowels /ɜ/ and /ə/

Other than /ə/, Rongpa has two contrastive central vowels, /ɜ/ and /ə/, as illustrated by the examples in (37):

(37) /vɜʰ/ 'tsampa'  /xpɜʰ/ 'pancreas'
     /vəʰ/ 'belly'  /xpəʰ/ 'potherb'

     /tsʰɜʰ/ 'goat'  /ʃɜʰ/ 'interest'
     /tsʰəʰ/→[tsʰz̩ʰ] 'wrinkle'  /ʃəʰ/→[ʃʒ̩ʰ] 'Selaginella'

     /xtʂɜʰ/ 'rheum officinale'  /xkɜʰ/ 'sound'
     /xtʂəʰ/→[xtʂɻ̩ʰ] 'waist'  /kə/ 'orientational prefix: leftward'

/ə/ has four allophones. After coronal fricatives and affricates, it is realized as syllabic voiced fricatives that are homorganic with the preceding consonant (in some transcription systems, these can be represented using symbols for "apical vowels"), and it is realized as [ə] elsewhere. Thus, the allophones of /ə/ and environments where they occur are summarized in Figure 3:

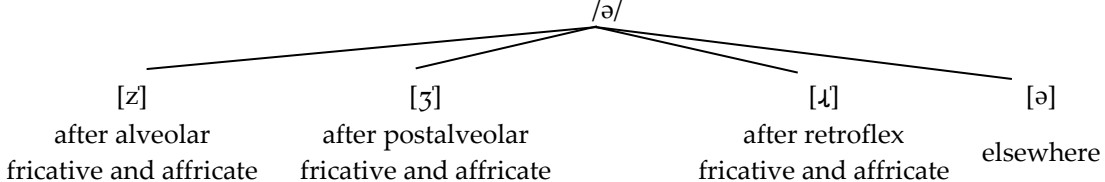

**Figure 3.** Allophones of the phoneme /ə/.

6.   Open front vowel /ɛ/

/ɛ/ (whose uvularized counterpart is /aᵸ/) is the only open plain vowel in the inventory (which can be distinctively characterized using [+low]) and can trigger vowel harmony (see Section 4.3.2 for detailed description).

*4.2. Uvularized Vowels*

The term "uvularization" is used less commonly in literature. Alternatively, the "velarization" is used to refer to a consonantal, rather than vocalic, feature (Ladefoged and Maddieson 1996, p. 365; Ladefoged and Johnson 2014, p. 245).

In recent years, more and more phonological and phonetic studies have been focusing on the phenomena of velarization/uvularization/pharyngealization in Sino-Tibetan languages (Sun 2000, 2004; Evans 2006a, 2006b; Lin et al. 2012; Sun and Evans 2013; Evans et al. 2016; Gong 2018, 2020; Chiu and Sun 2020). Such phenomena in previous studies are sometimes referred to as those of "tense" and "lax" vowels (Huang 1991a) or RTR (Gao 2015). In recent analyses, velarized or pharyngealized vowels are analyzed as a vocalic feature. In particular, "uvularized vowels" are detected and investigated in Tibeto-Burman languages such as Qiang (Sun and Evans 2013; Evans et al. 2016) and Minyag (Van Way 2018).

There are five uvularized vowels in Rongpa, as is schematized in Figure 4.

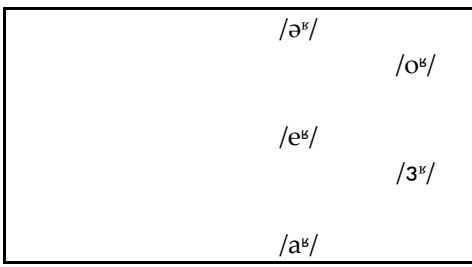

**Figure 4.** Rongpa uvularized vowels inventory.

1.     A phonemic analysis

In phonological terms, all the uvularized vowels are non-front vowels. Minimal sets are as follows:

(38)  /oʁ/      /xpoʁᴴ/ 'ice'                    /tsoʁᴴ/ 'bridge'
       /əʁ/      /xpəʁᴴ/ 'to blow: 1ᴘʟ'           /tsəʁᴴ/ 'ankle'
       /eʁ/      /xpeʁᴴ/ 'official; leader'       /tseʁᴸ tseʁᴴ/ 'be wet'
       /ɜʁ/      /xpɜʁᴴ/ 'goiter; pus'            /tsɜʁᴴ/ 'deer'
       /aʁ/      /xpaʁᴸ/ 'toad'                   /tsaʁᴴ/ 'fat [N]'

Each of the uvularized vowels contrasts with its plain counterpart; minimal pairs are:

(39)  /o/      /koᴸ/ 'porcupine'                        /ə/      /tʂʰəᴴ Nbuᴴ/ 'buttocks'
       /oʁ/     /koʁᴸ/ 'valley'                          /əʁ/     /tʂʰəʁᴴ Nbuᴴ/ 'ox horn'

       /e/      /xpeᴴ/ 'Tibetan incense'                 /ɜ/      /xpɜᴴ/ 'pancreas'
       /eʁ/     /xpeʁᴴ/ 'leader; official'               /ɜʁ/     /xpɜʁᴴ/ 'goiter; pus'

       /ɛ/      /kʰɛᴸ kʰɛᴴ/ 'to break up (the family)'
       /aʁ/     /kʰaʁᴸ kʰaʁᴴ/ 'be angry'

2.     Allophones of vowel /ɜʁ/

Impressionistically, the phonetic realization of /ɜʁ/ resembles the back open-mid vowel [ʌ] after coronal consonants (see examples in (40)) and [ɔ] after bilabial and velar consonants (see examples in (41)). These allophones appropriately reflect the influence of the gesture of uvularization (i.e., the retraction of the tongue root) on vowels. In other words, both [ʌ] and [ɔ] are endowed with the nature of uvularization.

(40)  /mɜʁᴸ/→[mɔᴸ] 'fire'
       /ɣɜʁᴸ/→[ʁɔᴸ] 'treetop'
(41)  /tsɜʁᴴ/→[tsʌᴴ] 'deer'
       /tʂɜʁᴴ/→[tʂʌᴴ] 'to squeeze'

Furthermore, the underlying nature of uvularization of [ʌ] and [ɔ] is also attested by the fact that both [ʌ] and [ɔ] can trigger uvularization harmony (see Section 4.3.1 for more detailed discussion).

The reason why /ɜʁ/ is paired with central open-mid plain vowel /ɜ/ lies in the parallel vowel alternation patterns of the verbs with nuclei /ɜ/ and /ɜʁ/. As shown in Table 17, /tsɜᴴ/ 'to milk (a cow)' and /xtsɜʁᴴ/ 'to pound (walnut)' show a parallel alternation pattern, that is, only the first-person-plural forms show a vowel alternation, while others remain unchanged. This alternation pattern only applies to verbs with nuclei /ɜ/ and /ɜʁ/ underlyingly.

**Table 17.** Vowel alternation patterns of /tsɜᴴ/ 'to milk (a cow)' and /xtsɜʁᴴ/ 'to pound (walnut)'.

| Subject | 1ꜱɢ | 1ᴘʟ | 2ꜱɢ | 2ᴘʟ | 3 |
|---|---|---|---|---|---|
| /tsɜᴴ/ 'to milk (a cow)' | tsɜᴴ | tsəᴴ | tsɜᴴ | tsɜᴴ | ptsɜᴴ |
| /xtsɜʁᴴ/ 'to pound (walnut)' | xtsɜʁᴴ | xtsəʁᴴ | xtsɜʁᴴ | xtsɜʁᴴ | xtsɜʁᴴ |

3.  Uvularized open vowel /aʁ/

/aʁ/ is the only open vowel among uvularized sets and is the uvularized counterpart of plain /ɛ/. The viability of pairing /aʁ/ with /ɛ/ can also be proved by a vowel harmony process, as will be discussed in Section 4.3.1.

4.  Phonological characteristics of plain and uvularized vowels

Vowel uvularization is essential in Rongpa phonology. Phonological evidence, as observed in phonotactic constrains and vowel harmony processes, can justify the application of the term "uvularization".

①  Phonotactic constraints

Phonotactic constraints show that Rongpa vowels can be divided into two natural classes. As can be seen in Table 18, in terms of phonetic realization, plain vowels do not occur with uvular initials, while uvularized vowels do not occur with velar initials, though both sets of vowels occur with labial and coronal initials.

**Table 18.** Distribution of plain and uvularized vowels in Rongpa. ("√" means the vowel can occur in that environment).

| Preceding Consonant \ Vowel Type | Plain Vowels | Uvularized Vowels |
|---|---|---|
| labial | √ | √ |
| alveolar | √ | √ |
| Postalveolar | √ | √ |
| retroflex | √ | √ |
| **velar** | √ | − |
| **uvular** | − | √ |

One plausible interpretation of the distribution in Table 18 is that velar and uvular initials are not phonemically distinct. As is stated above in Section 3.1, uvular consonants are allophones of velar consonants when followed by uvularized vowels.

②  Evidence from the first element /x, ɣ/ of the complex initials

The first element /x, ɣ/ assimilates to uvular fricative [χ, ʁ] when the nucleus of the syllable is a uvularized vowel[21], e.g.,

(42)  /**x**poʁH/→[**χ**poʁH] 'ice'
      /**x**ʧʰəʁH/→[**χ**ʧʰəʁH] 'dog'
      /**x**kaʁH/→[**χq**aʁH] 'to carve: 3'

Moreover, the uvularized vowel can also trigger progressive assimilation, causing the first element /x/ of the following to become [χ]. Compare the realization of the morpheme /xpeL/ "tree" in each example in (37).

(43)  /xtʂɛH gɛH **xpe**L/→[xtʂɛH gɛH **xpe**L] 'walnut tree'
      /lo**ʁ**H **xpe**L/→[lo**ʁ**H **χpe**L] 'tree'

The vowel of the next syllable is immune to such assimilation, which indicates that uvularization is not a suprasegmental feature that can permeate all syllables of a word.

③  Evidence from vowel harmony processes

Uvularization spreads leftwards from the stem to the prefix(es). If a verb root has a uvularized vowel as its nucleus, orientation prefixes attached to it can be fully uvularized (see detailed discussion later in Section 4.3.1). Consider the examples in (38), in which the orientational prefixes *kə-* 'ORT: leftward' and *ɣə-* 'ORT: rightward' become [qəʁ] and [ʁəʁ], respectively, because the underlying vowel of the verb root is uvularized.

(44)  /kə-xtə**ʁ**H/→[**qə**ʁH-xtəʁL] 'ORT: PFV-to ask: 1PL'
      /ɣə-bzə**ʁ**H/→[**ʁə**ʁH-βzəʁL] 'ORT: PFV -to pull: 1SG'

*4.3. Vowel Harmony*

Vowel harmony refers to assimilations among vowels that may be separated by consonants (Rose and Walker 2011, p. 251). Rongpa exhibits a rich array of vowel harmony processes in various domains: uvularization harmony (Section 4.3.1), height harmony (Section 4.3.2), and harmony on lip-roundedness (Section 4.3.3), as will be discussed in the following subsections.

4.3.1. Uvularization Harmony

Uvularization harmonizes leftward from stem to prefix. This process can be observed in various domains:

1.  Question marker $\varepsilon^H$-, non-past negative prefix *mε*-, prohibitive prefix *tε*-

When the vowel of the root is a plain vowel, the question marker, non-past negative, and prohibitive prefixes surface as [$\varepsilon^H$-], [mε-], and [tε-], respectively. On the other hand, when the nucleus of the verb root is a uvularized vowel, these prefixes become [aʁ$^H$-], [maʁ-], and [taʁ-]. Contrast the examples in (45) and (46).

(45)  /ε$^H$-dʒo$^H$/→[ε$^H$-dʒo$^H$] 'Q-to run: 2sɢ'
/mε-dʒo$^H$/→[mε$^H$-dʒo$^L$] 'ɴᴇɢ: ɴᴘsᴛ-to run: 1sɢ'
/tε$^H$-dʒo$^L$/→[tε$^H$-dʒo$^L$] 'ᴘʀᴏʜ-to run: 2sɢ'

(46)  /ε$^H$-xtsʒʁ$^H$/→[aʁ$^H$-χtsʒʁ$^H$] 'Q-to pound $_{(walnut)}$: 2sɢ'
/mε-xtsʒʁ$^H$/→[maʁ$^H$-χtsʒʁ$^L$] 'ɴᴇɢ: ɴᴘsᴛ-to pound $_{(walnut)}$: 1sɢ'
/tε-xtsʒʁ$^H$/→[taʁ$^H$-χtsʒʁ$^L$] 'ᴘʀᴏʜ-to pound $_{(walnut)}$: 2sɢ'

2.  Orientation prefix[22]

In Rongpa, the perfective aspect is encoded by adding an orientational prefix to the verb root. The orientation prefix harmonizes with the vowel of the verb root.

As already mentioned in Section 4.2, the vowel of the prefix is uvularized when it is attached to a verb root that has a uvularized vowel as its nucleus:

(47)  /tə-xtoʁ$^H$/→[təʁ$^H$-χtoʁ$^L$] 'ᴏʀᴛ:ᴘғᴠ-to rob: 1sɢ'
/lə-kʰəʁ$^L$/→[ləʁ$^L$-qʰəʁ$^H$] 'ᴏʀᴛ:ᴘғᴠ-to collect $_{(firewood)}$: 1ᴘʟ'
/kə-xmeʁ$^H$/→[qəʁ$^H$-χmeʁ$^L$] 'ᴏʀᴛ:ᴘғᴠ-to close $_{(eyes)}$: 2ᴘʟ'
/kə-xnʒʁ$^L$/→[qəʁ-χnʒʁ$^L$] 'ᴏʀᴛ:ᴘғᴠ-to light up: 1sɢ'
/lə-P-tʂʰaʁ$^H$/→[laʁ$^H$-ptʂʰaʁ$^L$] 'ᴏʀᴛ:ᴘғᴠ-to take apart: 3'

Note that, for orientation prefixes *kə*- and *ɣə*-, it is clear that, auditorily, not only the nucleus but also the onset of the prefix could be uvularized. In other words, the feature of uvularization spreads through the whole preceding syllable.[23]

(48)  /kə-xpəʁ$^H$/→[qəʁ$^H$-χpəʁ$^L$] 'ᴏʀᴛ: ᴘғᴠ-to blow: 1ᴘʟ'
/ɣə-xtʂʰəʁ$^L$/→[ʁəʁ$^L$-χtʂʰəʁ$^H$] 'ᴏʀᴛ: ᴘғᴠ-to untie: 1ᴘʟ'

3.  "*tə*- 'one' + classifier/quantifier" combinations

When the numeral *tə*- 'one' is followed by a classifier/quantifier, the vowel of *tə*- harmonizes with the vowel of the following syllable in terms of uvularization. Compare the examples in (49) and (50)

(49)  [tə$^H$ ɣu$^H$] 'a bundle (of wood)'
[tə$^L$ tsə$^H$] 'a piece (of bamboo)'
[tə$^L$ ʐε$^H$] 'a bowl (of rice)'

(50)  [təʁ$^L$ xsoʁ$^H$] 'a liǎng (in Chinese liǎng 两, a unit of weight that equals to 50 g)'
[təʁ$^L$ sʰʒʁ$^H$] 'a pail (of water)'
[təʁ$^L$ qʰaʁ$^H$] 'a package (of tobacco)'

4.3.2. Height Harmony

Vowel harmony on height systematically occurs on the orientation prefixes that are attached to the verb root. Specifically, when orientation prefix *tə*- 'ᴏʀ: unmarked' or *lə*- 'ᴏʀ:

downward' is attached to roots with [+low] nuclei /ɛ, aʁ/, the vowel of the prefix becomes [ɛ, aʁ] as well. See examples in (51)–(52). Note that examples in (52) exhibit both height and uvularization harmony.

(51)  /tə-dʒɛ^H/→[tɛ^H-dʒɛ^L] 'ORT:PFV-to run: 1PL'
      /lə-kʰɛ^H/→[lɛ^H-kʰɛ^L] 'ORT:PFV-to bask (under the sunshine): 1PL'

(52)  /tə-zaʁ^H/→[taʁ^H-zaʁ^L] 'ORT:PFV-to hit: 3'
      /lə-kʰwaʁ^L/→[laʁ^L-qʰwaʁ^H] 'ORT:PFV-to collect (firewood): 3'

Note that, however, orientation prefixes *kə-* 'OR: leftward' and *ɣə-* 'OR: rightward' do not show this height harmony:

(53)  /kə-xtɛ^L/→[kə^L-xtɛ^H] *[kɛ^H-xtɛ^L] 'ORT:PFV-to forge (iron): 1PL'
      /ɣə-dzaʁ^L/→[ʁə^L-dzaʁ^H] *[ʁa^L-dzaʁ^H] 'ORT:PFV-to thread (a needle): 3'

### 4.3.3. Harmony on Roundedness

Roundedness harmony is optional. For instance, it sporadically occurs on the prefix that is attached to the verb stem indicating first-person-singular form (the root vowel always alternates to /o/ and /oʁ/ with feature of [+round] when in 1st person singular form), as shown in examples (54)–(55). Note that examples in (55) exhibit both rounding and uvularization harmony.

(54)  /kə-dzo^H/→[ko^H-dzo^L]~[kə^H-dzo^L] 'ORT:PFV-to teach: 1SG'
      /ɣə-tʰo^H/→[ɣo^H-tʰo^L]~[ɣə^H-tʰo^L] 'ORT:PFV-to drink: 1SG'

(55)  /ɣə-dzoʁ^L/→[ʁoʁ^L-dzoʁ^H]~[ʁə^L-dzoʁ^H] 'ORT:PFV-to knead (needle): 1SG'
      /lə-roʁ^L/→[loʁ^L-roʁ^H]~[lə^L-roʁ^H] 'ORT:PFV-to laugh: 1SG'

## 5. Tone and Pitch Patterns

Choyul tones have been more or less described for several Choyul varieties by various scholars (Lu 1985; Wang 1991; Nishida 2008; Suzuki et al. n.d.; Suzuki and Wangmo 2018), though all these analyses are rather sketchy. In this section we tend to provide a more detailed, though not complete, description of the tonal system of Rongpa.

### 5.1. Tones in Monosyllabic Words

Monosyllabic words contrast two tones, /H/ and /L/. Impressionistically, some /H/ tone syllables can be realized on a high-level pitch, which is transcribed as [44] (see pitch trace in Figure 5a), while other /H/-tone syllables are realized on a high-falling contour, which is transcribed as [42] (see pitch trace in Figure 6a). Note that there is no contrast between [44] and [42]). On the other hand, /L/-tone syllables are realized on a low-rising contour in isolation, which is transcribed as [24] (see Figures 5b and 6b). Consider the minimal pairs in Table 19.

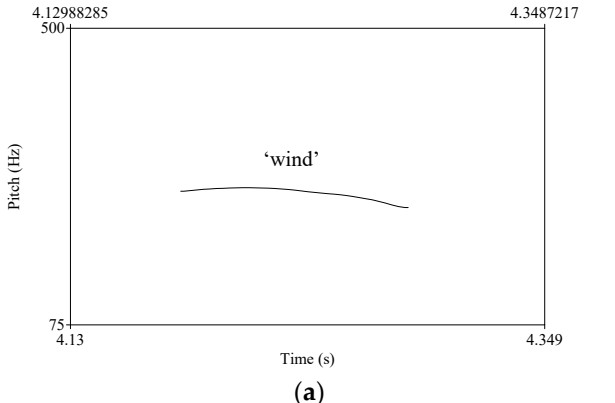

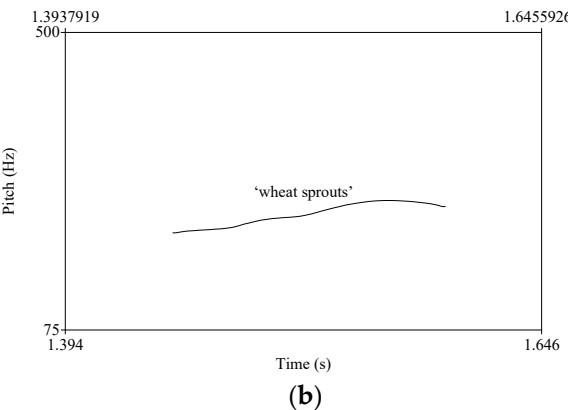

**Figure 5.** Pitch traces of /l̥aʁ^H/ 'wind' (**a**) vs. /l̥aʁ^L/ 'wheat sprouts' (**b**).

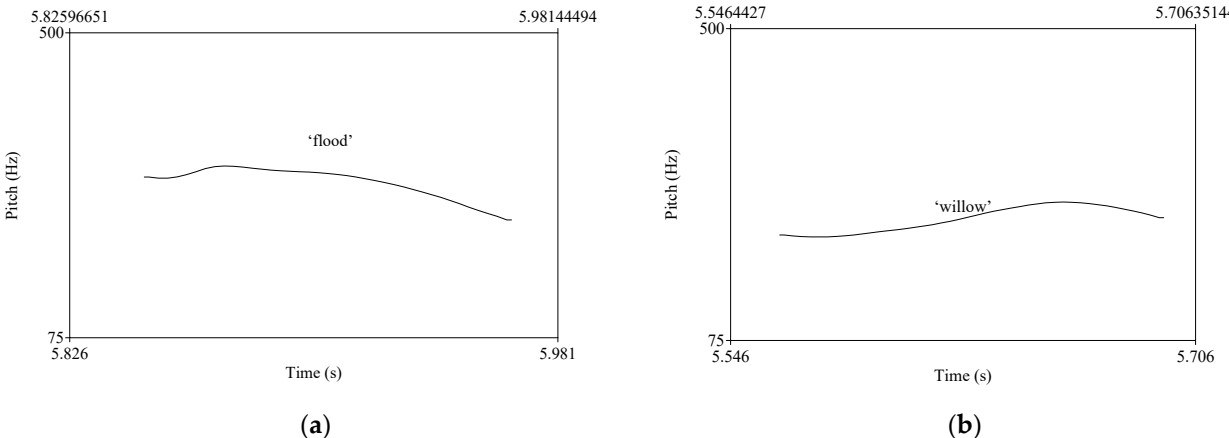

**Figure 6.** Pitch traces of /xtʂəˤᴴ/ 'flood' (**a**) vs. /xtʂəˤᴸ/ 'willow' (**b**).

**Table 19.** Rongpa tones on monosyllable words.

| Tone | Pitch Pattern | Examples | |
|------|---------------|----------|--|
| /H/ | [44]~[42] | /l̥aˤᴴ/→[l̥aˤ⁴⁴] 'wind' | /xtʂəˤᴴ/→[χtʂəˤ⁴²] 'flood' |
| /L/ | [24] | /l̥aˤᴸ/→[l̥aˤ²⁴] 'wheat sprouts' | /xtʂəˤᴸ/→[χtʂəˤ²⁴] 'willow' |

Figures 5 and 6 exhibit the pitch traces of the example shown in Praat (Boersma and Weenink 2018):

Rongpa tones are contrastive in all types of syllables.[24] The following minimal pairs in Table 20 demonstrate the binary contrast with all types of initials and rhymes:

**Table 20.** Rongpa tone minimal pairs on monosyllables.

| /H/ | /L/ |
|-----|-----|
| /toˤᴴ/ 'wing' | /toˤᴸ/ 'shell' |
| /pʰoᴴ/ 'to escape: 1SG' | /pʰoᴸ/ 'to lose (money): 1SG' |
| /goᴴ/ 'supper' | /goᴸ/ 'back (of the body)' |
| /xtaˤᴴ/ 'tiger' | /xtaˤᴸ/ 'wall (made by stone)' |
| /xtʂəˤᴴ/ 'flood' | /xtʂəˤᴸ/ 'willow' |
| /dʒiᴴ/ 'to run' | /dʒiᴸ/ 'scrotum' |
| /xuᴴ/ 'rain' | /xuᴸ/ 'yoghurt' |
| /sʰeᴴ/ 'mind' | /sʰeᴸ/ 'firewood' |
| /ɣoˤᴴ/ 'help [N]' | /ɣoˤᴸ/ 'madman' |
| /ɣʒəˤᴴ/ 'balcony' | /ɣʒəˤᴸ/ 'to mix: 1PL' |
| /l̥aˤᴴ/ 'wind' | /l̥aˤᴸ/ 'wheat sprouts' |
| /nɛᴴ/ 'owl' | /nɛᴸ/ 'fish' |

*5.2. Pitch Patterns in Disyllabic Words*

Rongpa disyllabic words exhibit four pitch patterns, as is summarized in Table 21:

**Table 21.** Pitch patterns in disyllabic words.

| Pitch Patterns | Examples | |
|:---:|:---:|:---:|
| [H-H] | [xte$^H$ re$^H$] 'fog' | [tʂʰə$^H$ mbu$^H$] 'buttocks' |
| [H-L] | [xnʒ$^H$ mbʒ$^L$] 'mouth' | [ɣoʶ$^H$ ʤʰi$^L$] 'sparrow' |
| [L-H] | [xlʒ$^L$ mnʒ$^H$] 'moon' | [pʰə$^L$ kʰɛ$^H$] 'stomach' |
| [LH-L] | [tsʒ$^{LH}$ nə$^L$] 'they' | [ge$^{LH}$ ni$^L$] 'be early' |

In our database, we cannot find minimal sets that contrast all the four patterns at once. We can only find minimal pairs that distinguish [L-H] from [H-H] (see examples in (56)) and those that contrast [L-H] and [H-L] (see examples in (57)). Note that there are no instances that distinguish [H-H] from [H-L].

(56) [L-H]  [ʃi$^L$ də$^H$] 'garlic'       [ɛ$^L$ kɛ$^H$] 'older brother'
     [H-L]  [ʃi$^H$ də$^L$] 'coral'       [ɛ$^H$ kɛ$^L$] 'tick (parasite)'

(57) [L-H]  [xko$^L$ xli$^H$] 'bone marrow'   [ʃi$^L$ mə$^H$] 'sand'
     [H-H]  [xko$^H$ xli$^H$] 'nine months'   [ʃi$^H$ mə$^H$] 'dried roasted barley'

The pattern [LH-L] is a moot point at this stage. The first syllable is produced with a complete low-rising contour and the second syllable is on a low pitch. The occurrence of this pattern may result from special morphological structuring. Throughout the whole database, there are three instances that are produced with this pattern. Note that in the first two examples in (58), =*nə* is probably a plural enclitic.

(58)  [pʒʶ$^{LH}$ nə$^L$] 'others'
      [tsʒ$^{LH}$ nə$^L$] 'they' (/tsʒ$^{LH}$/ 'he; she')
      [ge$^{LH}$ ni$^L$] 'be early'

### 5.3. Pitch Patterns in Verb Morphology

Monosyllabic verb roots also contrast two tones: /H/ and /L/. In general, a verb can take at most two prefixes, with a negative prefix slot located immediately following the orientational prefix: ᴏʀᴛ-ɴᴇɢ-sᴛᴇᴍ (∑). This section examines the pitch variations of perfective and imperative verb forms (Section 5.3.1), and negative verb forms (Section 5.3.2), which to an extent preserve the two-way contrast between /H/ and /L/.

#### 5.3.1. Perfective and Imperative Verb Forms

In Rongpa, a verb stem with an orientation prefix can either indicate the perfective aspect or imperative mood. Orientation prefixes are toneless in Rongpa, as their surface pitch realization is determined by the underlying tone value of the verb root. Take disyllabic verb forms as examples. If the verb root is /H/ toned, a perfective verb formed with this root is also /H/-toned and is realized as [H-L] if it is disyllabic when attached with an orientational prefix. A disyllabic perfective verb with a /L/-toned root, on the other hand, is realized as [L-H]. The tone patterns are fixed and do not result from spreading. Consider the examples in (59)–(60):

(59) Perfective: Underlying /H/→surface [H-L]
[kə$^H$-lo$^L$] (ᴏʀᴛ: ᴘꜰᴠ-∑) 'I have planted'
[ʁə$^ʁ$-l̥ɜ$^{ʁL}$] (ᴏʀᴛ: ᴘꜰᴠ-∑) 'I have herded $_{(sheeps)}$'
[rə$^ʁ$-pto$^{ʁL}$] (ᴏʀᴛ: ᴘꜰᴠ-∑) 'I have twined'
[lə$^H$-tsɜ$^L$] (ᴏʀᴛ: ᴘꜰᴠ-∑) 'I have milked $_{(the\ cow)}$'
[tə$^H$-s$^h$ɜ$^L$] (ᴏʀᴛ: ᴘꜰᴠ-∑) 'I have killed'

(60) Perfective: Underlying /L/→surface [L-H]:
[kə$^L$-xto$^H$] (ᴏʀᴛ: ᴘꜰᴠ-∑) 'I have pounded'
[ʁə$^ʁ$-χno$^{ʁH}$] (ᴏʀ: ᴘꜰᴠ-∑) 'I have peeled off'
[rə$^L$-nt$^h$o$^H$] (ᴏʀ: ᴘꜰᴠ-∑) 'I have held $_{(in\ hands)}$'
[lə$^ʁ$-qo$^{ʁH}$] (ᴏʀ: ᴘꜰᴠ-∑) 'I have picked'
[tə$^ʁ$-zo$^{ʁH}$] (ᴏʀ: ᴘꜰᴠ-∑) 'I have taken away'

Imperative verb forms in Rongpa are also composed of an orientation prefix plus a verb stem. The perfective and imperative verb forms of exactly the same /L/-toned verbs alternate between /L/ and /H/ tones in the two verb forms, Consider the examples in (61).

(61) Imperative: Underlying /H/→surface [H-L]:
[kə$^H$-xto$^L$] (ᴏʀᴛ: ɪᴍᴘ-∑) 'You pound!'
[ʁə$^ʁ$-χno$^{ʁL}$] (ᴏʀᴛ: ɪᴍᴘ-∑) 'You peel off!'
[rə$^H$-nt$^h$o$^L$] (ᴏʀᴛ: ɪᴍᴘ-∑) 'You hold!'
[lə$^ʁ$-qo$^{ʁL}$] (ᴏʀᴛ: ɪᴍᴘ-∑) 'You pick!'
[tə$^ʁ$-zo$^{ʁL}$] (ᴏʀᴛ: ɪᴍᴘ-∑) 'You take away!'

We would expect examples in (62) to have undergone the same polar alternation process in (61) by switching between the perfective and imperative forms. However, the imperative counterparts of the examples in (62) are actually realized as [H-H] rather than expected [L-H] (as the perfective /L/-toned pattern observed in (60)). A possible explanation is that [H-H] is another surface realization of /L/, specifically for inflected verbs. This hypothesis, however, requires further investigation in future research.

(62) Imperative: Underlying /L/→surface [H-H]:
[kə$^H$-lo$^H$] (ᴏʀᴛ: ɪᴍᴘ-∑) 'You plant!'
[ɣə$^H$-l̥ɜ$^{ʁH}$] (ᴏʀᴛ: ɪᴍᴘ-∑) 'You herd!'
[rə$^ʁ$-pto$^{ʁH}$] (ᴏʀᴛ: ɪᴍᴘ-∑) 'You twine!'
[lə$^H$-tsɜ$^H$] (ᴏʀᴛ: ɪᴍᴘ-∑) 'You milk!'
[tə$^H$-s$^h$ɜ$^H$] (ᴏʀᴛ: ɪᴍᴘ-∑) 'You kill!'

5.3.2. Negative Prefixes

There are three negative prefixes in Rongpa.

The non-past negative *mɛ-* is toneless and its surface pitch is determined by the underlying tone of the verb root. Compare the pitch pattern in (63)–(64):

(63) Underlying /H/→[H-L]:
[mɛ$^H$-lo$^L$] (ɴᴇɢ: ɴᴘsᴛ-∑) 'I don't plant'
[ma$^ʁ$-l̥ɜ$^{ʁL}$] (ɴᴇɢ: ɴᴘsᴛ-∑) 'I don't herd'
[ma$^ʁ$-pto$^{ʁL}$] (ɴᴇɢ: ɴᴘsᴛ-∑) 'I don't twine'
[mɛ$^H$-tsɜ$^L$] (ɴᴇɢ: ɴᴘsᴛ-∑) 'I don't milk'
[mɛ$^H$-s$^h$ɜ$^L$] (ɴᴇɢ: ɴᴘsᴛ-∑) 'I don't kill'

(64) Underlying /L/→[L-H]:
[mɛ$^L$-xto$^H$] (ɴᴇɢ: ɴᴘsᴛ-∑) 'I don't pound'
[ma$^ʁ$-χno$^{ʁH}$] (ɴᴇɢ: ɴᴘsᴛ-∑) 'I don't peel off'
[mɛ$^L$-nt$^h$o$^H$] (ɴᴇɢ: ɴᴘsᴛ-∑) 'I don't hold'
[ma$^ʁ$-qo$^{ʁH}$] (ɴᴇɢ: ɴᴘsᴛ-∑) 'I don't pick'
[ma$^ʁ$-zo$^{ʁH}$] (ɴᴇɢ: ɴᴘsᴛ-∑) 'I don't take away'

Past negative prefix *mə-* appears after the orientational prefix. Past negative verb forms surface on only one pitch pattern—[H-H-L], irrespective of its lexical tone, as examples in (65):

(65) Both underlying /H/ and /L/→[H-H-L]:
[kə$^H$-mə$^H$-lo$^L$] (ᴏʀ-ɴᴇɢ:ᴘsᴛ-∑) 'I haven't planted'
[kə$^H$-mə$^H$-s$^h$ʒ$^L$] (ᴏʀ-ɴᴇɢ:ᴘsᴛ-∑) 'I haven't killed'
[lə$^ʁH$-mə$^ʁH$-qo$^ʁL$] (ᴏʀ-ɴᴇɢ:ᴘsᴛ-∑) 'I haven't picked'
[lə$^ʁH$-mə$^ʁH$-zo$^ʁL$] (ᴏʀ-ɴᴇɢ:ᴘsᴛ-∑) 'I haven't taken away'

Prefix *tɛ-* attaches to the verb stem to express a prohibitive mood. A prohibitive verb form with a /H/-toned stem is produced on the [H-L] pattern, while its /L/-toned counterpart is realized as [LH-L], compare pitch patterns in (66)–(67):

(66) Underlying /H/→[H-L]:
[tɛ$^H$-lo$^L$] (ᴘʀᴏʜ-∑) 'Don't plant!'
[ta$^ʁH$-ḻʒ$^ʁL$] (ᴘʀᴏʜ-∑) 'Don't herd!'
[ta$^ʁH$-pto$^ʁL$] (ᴘʀᴏʜ-∑) 'Don't twine!'
[tɛ$^H$-ts3$^L$] (ᴘʀᴏʜ-∑) 'Don't milk!'
[tɛ$^H$-s$^h$ʒ$^L$] (ᴘʀᴏʜ-∑) 'Don't kill!'

(67) Underlying /L/→[LH-L]:
[tɛ$^{LH}$-xto$^L$] (ᴘʀᴏʜ-∑) 'Don't pound!'
[ta$^{ʁLH}$-χno$^ʁL$] (ᴘʀᴏʜ-∑) 'Don't peel off!'
[tɛ$^{LH}$-nt$^h$o$^L$] (ᴘʀᴏʜ-∑) 'Don't hold!'
[ta$^{ʁLH}$-qo$^ʁL$] (ᴘʀᴏʜ-∑) 'Don't pick!'
[ta$^{ʁLH}$-zo$^ʁL$] (ᴘʀᴏʜ-∑) 'Don't take away!'

Pitch patterns as observed in verb morphology are summarized in Table 22:

**Table 22.** Pitch patterns in Rongpa verb morphology.

| ∑ | Perfective ᴏʀᴛ: ᴘꜰᴠ-∑ | Imperative ᴏʀᴛ: ɪᴍᴘ-∑ | Non-past Negative ɴᴇɢ: ɴᴘsᴛ-∑ | Past Negative ᴏʀᴛ-ɴᴇɢ: ᴘsᴛ-∑ | Prohibitive ᴘʀᴏʜ-∑ |
|---|---|---|---|---|---|
| /H/ | [H-L] | [H-H] | [H-L] | [H-H-L] | [H-L] |
| /L/ | [L-H] | [H-L] | [L-H] | | [LH-L] |

## 6. Concluding Remarks

This paper presents a preliminary analysis of phonology of Rongpa Choyul, an understudied Sino-Tibetan language, based on firsthand fieldwork data. The main findings are as follows:

1. Rongpa has a substantial phonemic inventory, which comprises 43 simple consonant initials, 84 consonant clusters as complex initials, 13 vowels and 2 contrastive tones. All the contrasts in this paper are proved by (near) minimal pairs/sets.
2. Regarding consonant inventory, affricates and fricatives in Rongpa show a three-way contrast in their places of articulation (i.e., dental, postalveolar, and retroflex). Coronal fricatives also show three-way contrast in their manner of articulation (i.e., voiceless unaspirated, voiceless aspirated, and voiced). The contrast of fricative aspiration is still remaining even in consonant clusters.
3. Rongpa exhibits a complex vocalic system. The phonemic contrast between plain vowels and uvularized vowels is attested. Uvularization is a conspicuous and indispensable vowel feature, which plays an essential role in phonotactic rules and vowel harmony process.
4. As for word prosody, two contrastive tones in monosyllabic words, /H/ and /L/, are used to distinguish lexical meanings, while disyllabic words mainly exhibit four pitch patterns: [H-H], [H-L], [L-H], and [LH-L], among which [H-H] and [H-L] are not con-

trastive. Pitch patterns in verb constructions are also examined, and polar alterna­tions are employed to distinguish different grammatical categories such as perfective and imperative.

The findings and analyses as presented in this paper could form a solid foundation for future research. The physiological and acoustic mechanism of uvularization in Choyul are important directions for further studies. Description of morphosyntax and comparative studies with other varieties of Choyul is of great value in understanding the phonology of Rongpa.

**Funding:** This research received no external funding.

**Institutional Review Board Statement:** Not applicable.

**Informed Consent Statement:** Not applicable.

**Data Availability Statement:** Not applicable.

**Conflicts of Interest:** The author declares no conflict of interest.

## Notes

1    Tuanjie Township (团结乡) has been renamed as Gara Town (ᴺᴿ 呷拉镇) since 1979 (c.f., (Yǎjiāng Xiànzhì Biānzuǎn Wěiyuánhuì (雅江县志编纂委员会) 2000, p. 39)).

2    Wang (1991, p. 46) mentioned that: "Speakers of Choyul in Nyagrong County refer to their language as [tɕho⁵⁵ kɛ⁵⁵], and the regions where [tɕho⁵⁵ kɛ⁵⁵] is spoken, including places belonging to Nyagchu and Lithang County, are collectively called [tɕho⁵⁵ y⁵⁵]. If [kɛ⁵⁵] and [y⁵⁵] were borrowed from Tibetan <skad> and <yul>, [tɕho⁵⁵ kɛ⁵⁵] and [tɕho⁵⁵ y⁵⁵] can be labeled as 'què (却) language' and 'què (却) region', respectively. However, the origin and meaning of 'què (却)' still remain ambiguous".

3    In previous literature, Choyul has been alternatively refer to "Zhaba" in various cases. Sun (1983, pp. 155–63; 2013, pp. 151–63) uses data of "Zhaba" collected by Shaozun Lu. Lu (1985) also uses the term "Zhaba (扎巴)" to refer to the Choyul variety that he studied (i.e., Tuanjie (团结) variety spoken in Nyaychu County). However, just as Huang (1991b, p. 65) points out, the language studied by Lu (1985) is actually a variety of Choyul, which is distinct from the bona fide Zhaba. Sun (2001, p. 158; 2016, p. 6) also admitted that Choyul had been mislabled as "Zhaba", though he misuses the term "Zhaba" again in Sun (2013, pp. 151–63).

4    According to Sun (2001), the notion "Qiangic" emerged in the early 1960s, constituting only three languages: Qiang, Pumi and Rgyalrong. It was expanded since the late 1970s, when a number of seemingly related languages (including Tangut) were added to the branch. Specifically, Sun (2001, p. 160) divided the said languages into two branches, subsuming Choyul and Zhaba into the Southern branch. Meanwhile, Sun noticed that the two languages also show diagnositic characteristics of both Northern and Southern branches, so he suggested that they could be in an "intermediate genetic position". After that, in Sun (2016, p. 4), he proposed a three-way division of "Qiangic" into: (1) the Northern branch, which comprises Rgyalrong (嘉戎), Ergong (尔龚) and Lavrung (拉坞戎); (2) the Central branch, consituting Zhaba (扎巴), Choyul (却域), Qiang (羌), Tangut (西夏), Muya (木雅) and Pumi (普米); and (3) the Southern branch, which includes Shixing (史兴), Namuyi (纳木义), Ersu (尔苏) and Guiqiong (贵琼).

5    /N/ stands for an archiphonemic nasal that is homorganic to the following consonant. See Section 3.2.2 for detailed discussion.

6    In this paper, tones/pitch patterns are marked by superscripted "H" and "L" after each syllable.

7    Abbreviations "1sɢ, 1ᴘʟ, 2sɢ, 2ᴘʟ, 3" are used to indicate the person and number of the subject that the verb alters for. Otherwise, the gloss of verbs without such specifications are underlying verb forms (without any vowel alternations). Other abbreviations occur in this paper include: 1 = first person, 2 = second person, 3 = third person, ɪᴍᴘ = imperative, ɴᴇɢ = negative, ɴᴘsᴛ = non-past, ᴏʀᴛ = orientational prefix, ᴘꜰᴠ = perfective, ᴘʟ = plural, ᴘʀᴏʜ = prohibitve, ᴘsᴛ = past, Q = question marker, sɢ = singular.

8    Note that the place of articulation of the consonants in these minimal pairs is not conditioned by the following vowel or the tone.

9    This distribution can be clearly observed when the verb /niᴸ/ 'to tell' undergoes vocalic alternation to conjugate for the person and number of the subject. When it comes to conjugation for a second-person-singular subject, the vowel alters to [ə] and the initial nasal is realized as [n]; while for first person singular and plural and second-person plural subjects, the vowels can trigger palatalization. Thus, the nasal initial is realized as [ɲ]. Consider the verb forms in this table:

| Subject | 1sɢ | 1ᴘʟ | 2sɢ | 2ᴘʟ |
|---|---|---|---|---|
| /niᴸ/ 'to tell' | [ɲoᴸ] | [ɲɛᴸ] | [nəᴸ] | [ɲeᴸ] |

10    Probably borrowed from Tibetan ꜱ <sna>.

11    Probably borrowed from Tibetan ꜱ <rna>.

[12] Another consultant of ours, Zhengna (正娜), who is a cousin of our language teacher, and who worked with Prof. You-Jing Lin in 2019, tends to pronounce the first elements /x/ and /ɣ/ more emphatically than our language teacher. She produces the pharyngeal sound in question with more force even when C1 is voiceless /x/. In her speech, C1 = /x/ is usually produced as a voiceless pharyngeal fricative [ħ], with a voiceless flap [ɾ̥] between C1 and C2, e.g., /ɣni$^H$/→[ʕɾni$^H$] 'ear'; /xni$^H$/→[ħɾ̥ni$^H$] 'nose'.

[13] Unlike in Japhug Rgyalrong (Jacques 2021, p. 48) and Thebo Tibetan (Lin 2018, p. 29), there is no solid evidence for treating the prenasalized voiced stops and affricates as single phonemes in Rongpa.

[14] The capitalized *P-* here represents an archiphonemic prefix that indexes the third-person subject. It has 5 allomorphs:

- Regular allmorphs: [p-], [ɸ-] and [β-] are in compementary distribution
  ① Voiceless unaspirated unreleased stop [p-]: When the initial of the verb root is a voiceless stop or affricate (featured as [-voiced, -continuant]), e.g., /P-tʰe$^H$/→[ptʰe$^H$] 'he drinks'; /P-ʧʰi$^H$/→[pʧʰi$^H$] 'he eats';
  ② Voiceless fricative [ɸ-]: When the initial of the verb root is a voiceless fricative or voiceless liquid (featured as [-voiced, +continuant]), e.g., /P-sʰʒ$^H$/→[ɸsʰʒ$^H$] 'he kills; /P-lʒ$^ʁH$/→[ɸlʒ$^ʁH$] 'he herds'; Moreover, *P-* is also realized as [ɸ-] before voiceless nasal, as if the voiceless nasals in Rongpa are also [+continuant], e.g., /P-n̥o$^H$/→[ɸn̥o$^H$] 'he stir fries'.
  ③ Voiced fricative [β-]: When the initial of verb root is a voiced fricative, or liquid (featured as [+voiced,+continuant]), e.g., /**P-**ʒo$^L$/→[βʒo$^L$] 'he learns'; /**P-**le$^H$/→[βle$^H$] 'he plants';

  If we observe the distribution of these allomorphs, we can conclude that the prefix *P-* assimilates with the manner of articulation of the following simple consonant of the verb root, which can be formulated using this phonological rule:

$$/P-/ \rightarrow \begin{bmatrix} \alpha \text{ continuant} \\ \alpha \text{ voice} \end{bmatrix} / \underline{\quad} \begin{bmatrix} \alpha \text{ continuant} \\ \alpha \text{ voice} \end{bmatrix}$$

  Note that *P-* will not realised as a voiced bilaibial stop [b-] before verb roots with voiced stops or affricates, e.g., /P-di$^L$/→[di$^L$] *[bdi$^L$] 'He accumulates'; /P-ʤi$^H$/→[ʤi$^H$] *[bʤi$^H$] 'He runs'.

- Irregualr allomorphs:
  ④ Nasal [m-]: When the initial of the verb root is a voiced nasal [n], the prefix *P-* is totally assimilates to [m-], e.g., /P-ni$^H$/→[mni$^H$] 'He hires/employes'; When /N/ is the first element of the complex initial, it coalesces with prefix *P-* into [m], e.g., /P-Ntʰe$^H$/→[mtʰe$^H$] 'He sings'; /P-Ndzə$^ʁL$/→[mdzə$^ʁL$] 'He arrests';
  ⑤ Glide [w]: When C is a velar consonant and the vowel of the root is uvularized /a$^ʁ$/, an insertion of glide [w] is applied: /P-kʰa$^ʁL$/→[qʰwa$^ʁL$] 'He picks';

All these allomorphs of *P-* show its [+labial, +round] nature.

[15] Note that in [loŋ$^L$ tʂʰə$^H$] 'food steamer' and [toŋ$^L$ tʂʰa$^ʁH$ tə$^L$ ro$^L$] 'one thousand', *[ŋtʂʰ] is not a legal cluster in Rongpa. Thus, [ŋ] is analysed as the coda of the preceding syllable.

[16] This is a unit of length which is a synonym for Chinese "一拃 (zhǎ)". Interestingly, Rongpa further distinguishes /Ndzə$^ʁH$/, which refers to the length between the thumb and forefinger, and /ptʂʰi$^H$/, which refers to the length between thumb and the middle finger.

[17] Note that the cluster /pl/→[ɸl] is only observed in three instances in our database, namely: /plɛ$^L$/→[ɸlɛ$^L$] 'to plait one's hair' (probably borrowed from Tibetan ⦿⦿⦿ <blas>); /pla$^H$/→[ɸlə$^H$] 'to unbraid one's hair; to spread out'; and /ple$^H$/→[ɸle$^H$] 'wooden plate'. Both /plɛ$^L$/ 'to plait one's hair' and /plə$^H$/ 'to unbraid one's hair; to spread out' undergoes the same morphological process involving vowel alternations to index the person and number of the subject like native Rongpa verbs:

| Subject | 1SG | 1PL | 2SG | 2PL | 3 |
|---|---|---|---|---|---|
| /plɛ$^L$/ 'to plait (one's hair)' | [ɸlo$^L$] | [ɸli$^L$] | [ɸlo$^L$] | [ɸle$^L$] | [ɸlɛ$^L$] |
| /plə$^H$/ 'to unbraid (one's hair); to spread out' | [ɸlə$^H$] | [ɸli$^H$] | [ɸlə$^H$] | [ɸlə$^H$] | [ɸlə$^H$] |

[18] The marginal status of "[b] + voiced stop/affricate" clusters is also justified by the fact that the third-person prefix *P-* is not realized as predicted [b] when it attaches to the verb roots with voiced stop or affricate initials (see footnote 14 for detaied discussion of the third-person prefix *P-*), e.g., /P-di$^L$/→[di$^L$] *[bdi$^L$] 'He accumulates'; /P-ʤi$^H$/→[ʤi$^H$] *[bʤi$^H$] 'He runs'; /P-dʐ̩$^H$/→[dʐ̩$^H$] *[bdʐ̩$^H$] 'He queues'.

[19] Probably borrowed from Tibetan ⦿⦿⦿ <btsa'> 'rust'.

[20] There is only one example of /i/ after labial onset in my current lexicon: /mi$^H$/→[mi$^H$] 'wound'(probably etymologically from Tibetan ⦿ <rma>).

[21] This phenomenon of assimilation can also seen as the uvularization of the whole syllable.

[22] The basic orientation system in Rongpa is shown in the following table in forms of prefixes:

| Subsystem | Oppositions | |
|---|---|---|
| vertical | *rə-*<br>'upward' | *lə-*<br>'downward' |
| solar & zone | *kə-* 'westward/inward' | *ɣə-* 'eastward/outward' |
| neutral | *tə-*<br>'unmarked' | |

These prefixes code both orientation and perfectivity. When prefixed with a motion verb, the orientational connotation of the prefix is explicitly specified. While in other cases, the selection of prefixes is dependent on the verb's inherent lexical semantics or can be totally conventionalized.

[23] The fact that the whole syllable is uvularized was also observed in Puxi Horpa (Lin et al. 2012, p. 193) and Mawo Qiang (Sun and Evans 2013, p. 141).

[24] Some Tibetic languages exhibits a partial tone system, in which the tones are only contrastive in restricted circumstances. In Thebo (迭部), for example, tones are only contrastive when the syllable bears sonorant or voiceless unaspirated initials (Lin 2014), while this is not the case in Rongpa.

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
