# Peer review of "A Phonological Study of Rongpa Choyul"

_languages, doi:10.3390/languages8020133_

Round 1

Reviewer 1 Report

Article is very interesting, but in places I feel like the analysis could be taken further. For example, -w- has a very limited distribution (only after velars), but it is also obligatory before schwas (as subphonemic). This is a fine description of facts, but I wonder, for examples whether [u] could be analyzed as /wi/ and whether this would not improve the analysis of both the glide and the vowels.

p. 4, the contrasts are in some cases not minimal, so I find myself wondering, e.g., whether z might just be an automatically voiced version of s between vowels. Where a three way minimal contrast cannot be found, the authors should rely on a pair of pairs.

Particularly in the cases of minimal contrasts, like f, I want to know more about the field conditions. How many distinct tokens of the word were collected? Was the speaker asked if the same word would be acceptable with phi, etc?

The palatalization before /o/ is interesting. Is this attested in other languages?

p. 9, one thing I find annoying about work on Qiangic and Rgyalrongic languages is that people can't decide whether they use phonemes or archiphonemes. If n- and ng- are distinct phonemes, then according to standard practice they are also distinct phonemes even where they never contrast. But N- is an archiphoneme. There are other parts of the language's phonology that could be approached with archiphonemes, but aren't. It is fine to stick with tradition here, perhaps best to do so, but it needs to be done more clearly and explicitly, for example by admitting that N is an archiphoneme, citing Trubetskoy, etc. saying why you are doing it. The first time someone sees a capital N they will think it is a uvular, so this all needs more explicit flagging.

Those charts that use parentheses and tildes need to say in the caption what these mean. In standard theory, one minimal contrast is enough to make something a phoneme (confusion vs Confucian) in all contexts. So, generally speaking, I don't approve of these things in parentheses. A phoneme is categorical. For something to be 'sort of a phoneme' makes no sense.

p. 12, explain more clearly why you decide on beta and phi as the underlying forms rather than b and p.

p. 13, since -w- only occurs after velars, one might as well say there are labiovelar simple consonants, no? 

The discussion of how the uvularized vowels pair with the non uvularized ones is very confusing. I think there is a typo somewhere. Please check and expand this section. Also, I would write both members of a pair with the same symbol, for the sake of clarity.

We are given some phonetic information for tone, but some for uvularization would also be nice. Also, we need more information about how the Praat files were processed. Was there are carrying phrase? How many tokens were used? etc.

There are about a dozen places where there are problems with English, some of which compromise intelligibilty. These must be checked.

Author Response

Dear reviewer:

I am very grateful to you for reviewing the paper so carefully and addressing incisive comments. I have carefully considered the suggestions and made some changes in the manuscript. Please find the attachment of my responses to reviewers' comments. 
